# Muscle-derived GDF15 drives diurnal anorexia and systemic metabolic remodeling during mitochondrial stress

Mario Ost[1,*,†,‡,§] (iD), Carla Igual Gil[1,2,†], Verena Coleman[1,2], Susanne Keipert[3], Sotirios Efstathiou[1], Veronika Vidic[1], Miriam Weyers[1] & Susanne Klaus[1,2,‡,**] (iD)

## Abstract

Mitochondrial dysfunction promotes metabolic stress responses in a cell-autonomous as well as organismal manner. The wasting hormone growth differentiation factor 15 (GDF15) is recognized as a biomarker of mitochondrial disorders, but its pathophysiological function remains elusive. To test the hypothesis that GDF15 is fundamental to the metabolic stress response during mitochondrial dysfunction, we investigated transgenic mice (*Ucp1*-TG) with compromised muscle-specific mitochondrial OXPHOS capacity via respiratory uncoupling. *Ucp1*-TG mice show a skeletal muscle-specific induction and diurnal variation of GDF15 as a myokine. Remarkably, genetic loss of GDF15 in *Ucp1*-TG mice does not affect muscle wasting or transcriptional cell-autonomous stress response but promotes a progressive increase in body fat mass. Furthermore, muscle mitochondrial stress-induced systemic metabolic flexibility, insulin sensitivity, and white adipose tissue browning are fully abolished in the absence of GDF15. Mechanistically, we uncovered a GDF15-dependent daytime-restricted anorexia, whereas GDF15 is unable to suppress food intake at night. Altogether, our evidence suggests a novel diurnal action and key pathophysiological role of mitochondrial stress-induced GDF15 in the regulation of systemic energy metabolism.

**Keywords** anorexia; GDF15; integrated stress response; mitochondrial dysfunction; muscle wasting
**Subject Categories** Metabolism; Molecular Biology of Disease

## Introduction

Mitochondria are essential organelles for cellular ATP production through oxidative phosphorylation (OXPHOS) as well as the regulation of cellular physiology [1]. Moreover, mitochondrial function is crucial for integration and generation of metabolic signals via interaction with other cellular organelles such as the endoplasmic reticulum [2]. Mitochondrial diseases are one of the most common types of inherited metabolic disorders that can manifest at different stages of age and in any organ with strong variability in clinical features [3]. Notably, the first human mitochondrial disease, Luft's disease, was discovered 60 years ago [4], with clinical presentation of skeletal muscle atrophy due to increased muscle-specific mitochondrial respiratory uncoupling, energy depletion, and systemic hypermetabolism [5]. Thereby, Luft and colleagues first described how a tissue-specific compromised mitochondrial function governs systemic energy homeostasis. Currently, our understanding of the mechanisms that promote the pathophysiological consequences of mitochondrial dysfunction still remains limited. Mounting evidence indicates that the impairment of mitochondrial OXPHOS is underlying the induction of an integrated stress response (ISR) that involves both cell-autonomous signaling pathways [6–8] as well as the induction of cytokines which potentially affect systemic metabolism in an cell-non-autonomous fashion [9,10]. However, crucial metabolic mediators guiding the endocrine crosstalk during muscle mitochondrial OXPHOS deficiency are poorly described.

In the last decade, the wasting hormone growth differentiation factor 15 (GDF15, also named macrophage inhibitory cytokine-1 [MIC-1], placental bone morphogenetic protein [PLAB], placental transforming growth factor-β [PTGF-beta], prostate derived factor [PDF], or non-steroidal anti-inflammatory drugs-activated gene [NAG-1]) gained momentum as a biomarker of cancer cachexia, cellular stress, mitochondrial dysfunction, and aging [11–13]. GDF15 belongs to the transforming growth factor beta superfamily

1  Department of Physiology of Energy Metabolism, German Institute of Human Nutrition Potsdam-Rehbrücke, Nuthetal, Germany
2  Institute of Nutritional Science, University of Potsdam, Nuthetal, Germany
3  Department of Molecular Biosciences, The Wenner-Gren Institute, Stockholm University, Stockholm, Sweden
   *Corresponding author. Tel: +49 341 97 15051; E-mail: mario.ost@dife.de
   **Corresponding author. Tel: +49 33200 88 2326; E-mail: klaus@dife.de
   †These authors contributed equally to this work as first authors
   ‡These authors contributed equally to this work as senior authors
   §Present address: Department of Neuropathology, Universitary Hospital Leipzig, Leipzig, Germany

and was first identified 1997 in activated macrophages [14] as well as a secretory protein with high expression in the placenta [15]. Circulating GDF15 is highly elevated in patients with cancer and severe anorexia [16], chronic inflammation [17], and pediatric heart disorder [18], as well as in patients with mitochondrial diseases [19–22]. By now, elevated skeletal muscle expression of GDF15 has been documented under conditions of muscle-specific mitochondrial stress in humans and different mouse models [7,20,23–25]. With regard to systemic energy homeostasis, a potent anorectic action of endogenous GDF15 was initially described in a mouse model of cancer and cachexia wasting syndrome [26] and very recently for exogenously administrated GDF15 in mice [27–31]. However, the biological function of GDF15 as muscle-secreted cytokine (also termed myokine) in response to mitochondrial dysfunction remains elusive.

Here, we propose a crucial role for endogenous GDF15 as a metabolic mediator in facilitating muscle mitochondrial stress-induced systemic metabolic remodeling. We report that GDF15 as a myokine is negligible for muscle mitochondrial function, cell-autonomous ISR induction, and mitochondrial stress-driven skeletal muscle atrophy. On the other hand, we uncovered that genetic ablation of GDF15 during muscle mitochondrial dysfunction prevented a daytime-restricted anorexia, promotes a robust fat mass expansion, and abolishes white adipose tissue browning and insulin sensitivity. Based on our results, muscle-derived GDF15 emerges as a critical regulator of diurnal energy balance which is instrumental for driving a systemic metabolic response during mitochondrial dysfunction.

# Results

## Muscle mitochondrial stress promotes GDF15 as a myokine

Using transgenic mice with compromised skeletal muscle-specific mitochondrial OXPHOS capacity via respiratory uncoupling (HSA-*Ucp1*-TG [TG]) [32,33], we explored the potential induction of GDF15 as a myokine (Fig 1A). First, using high-resolution respirometry of permeabilized muscle fibers, we confirmed the increased uncoupled (LEAK) respiration in soleus (SOL, slow-oxidative fiber type) and extensor digitorum longus (EDL, fast-glycolytic fiber type) of TG mice compared to wild-type (WT) controls (Fig 1B). Next, we validated the induction of muscle ISR via elevated gene expression of *Atf4*, *Atf5*, *Atf6*, and *Chop* (Fig 1C) and phosphorylation of eukaryotic translation initiation factor 2 alpha (eIF2α) (Fig 1D and E) in TG mice. Recently, a CHOP-dependent induction of GDF15 during muscle ISR was described [23]. To test for *Gdf15* induction in TG mice, we performed a multi-tissue transcriptomic profiling. In line with the HSA-driven skeletal muscle-specific UCP1 transgene [32], we found a strong induction of *Gdf15* exclusively in skeletal muscles of TG mice, with the highest induction observed in mixed and predominantly fast-glycolytic fiber type muscle (EDL, tibialis anterior [TA], gastrocnemius [Gastroc], quadriceps [Quad]) and a lower induction in oxidative fiber type muscles (SOL, diaphragm, esophagus) of TG animals (Fig 1F and G). Importantly, *Gdf15* expression was not affected in the heart, as well as in non-muscle tissues such as liver, kidney, spleen, lung, or different adipose tissue depots of TG mice. Accordingly, skeletal muscle GDF15 protein expression (Fig 1H) and *ex vivo* secretion from soleus or EDL

muscles were induced in TG mice only (Fig 1I). Plasma concentrations of circulating GDF15 levels were strongly increased in TG mice independent of sex, low or high caloric diet, or age (Fig 1J–L). To confirm the induction of GDF15 by mitochondrial uncoupling in muscle cells *in vitro*, we treated immortalized mouse C2C12 myocytes for 5 h with the chemical mitochondrial uncoupler FCCP (Fig 1M) and revealed a dose-dependent mRNA induction of *Gdf15* and ISR markers (Fig 1N). Altogether, these findings validate our genetic mouse model of muscle mitochondrial stress-induced ISR induction and skeletal muscle-derived GDF15 secretion.

## GDF15-independent muscle wasting and mitochondrial integrated stress response

Given the robust induction of GDF15 during muscle mitochondrial uncoupling and the fact that the pathophysiological role of ISR-induced GDF15 as a myokine is unknown, we crossbred TG and whole-body *Gdf15*-knockout (KO) [34] mice and performed a comprehensive *in vivo* phenotypic, metabolic, and molecular profiling in WT, KO, TG, and *Ucp1*-TGx*Gdf15*-KO (TGxKO) animals. All mouse experiments were accomplished under standard chow feeding conditions. The genotypes were determined by PCR (Fig 2A) and further confirmed on mRNA expression level (Fig 2B). We further confirmed elevated plasma GDF15 levels in TG mice, which were undetectable in age-matched KO and TGxKO mice (Figs 2C and EV1A).

Consistent with previous findings [35], male KO mice were indistinguishable in appearance and skeletal muscle phenotype from age-matched WT controls (Fig EV1B–H), suggesting that GDF15 does not play a major role as a physiological regulator of skeletal muscle function and body weight in healthy animals. Next, we sought to evaluate the muscle cell-autonomous relevance of GDF15 induction for skeletal muscle during compromised mitochondrial function. In both TG and TGxKO mice, voluntary wheel running (VWR) activity was not impaired (Fig 2D) whereas muscle grip strength was dramatically reduced compared to WT controls (Fig 2E). As reported previously [7], we found a 50% decrease of quadriceps and gastrocnemius muscle-to-lean body mass ratio in TG mice, suggesting a substantial wasting of skeletal muscle tissue (Fig 2F). This skeletal muscle atrophy was unaffected in TGxKO mice (Figs 2F and EV2A). In addition, morphological analyses showed no presence of severe structural alterations but a 40% reduction of the cross-sectional area of myofibers in both TG and TGxKO mice at 20 weeks (Fig EV2B and C) and up to 95 weeks of age (Fig 2G and H). Immunoblotting analyses confirmed a muscle fiber switch toward slow-oxidative type [36], and activation of AMP-activated protein kinase (AMPK) as well as a preserved protein expression of mitochondrial OXPHOS complexes in muscles of TG and TGxKO mice (Fig EV2D–I). Furthermore, characterization of skeletal muscle mitochondrial respiratory capacity (Fig 2I) displayed no difference between TG and TGxKO mice. Both oxidative soleus and glycolytic EDL muscles showed higher uncoupled respiration (Fig 2J and K), consequently causing a reduction in total OXPHOS capacity independent of GDF15 action (Fig 2L). Mitochondrial respiratory function and OXPHOS protein expression were unaffected in KO controls (Fig EV1I,J).

We next evaluated skeletal muscle mitochondrial stress-induced ISR induction. Here, we found that gene expression of ISR components *Atf4*, *Atf6*, and *Chop* (Fig 2M) and phosphorylation of eIF2a protein (Figs 2N and EV2J) remained highly induced in TGxKO

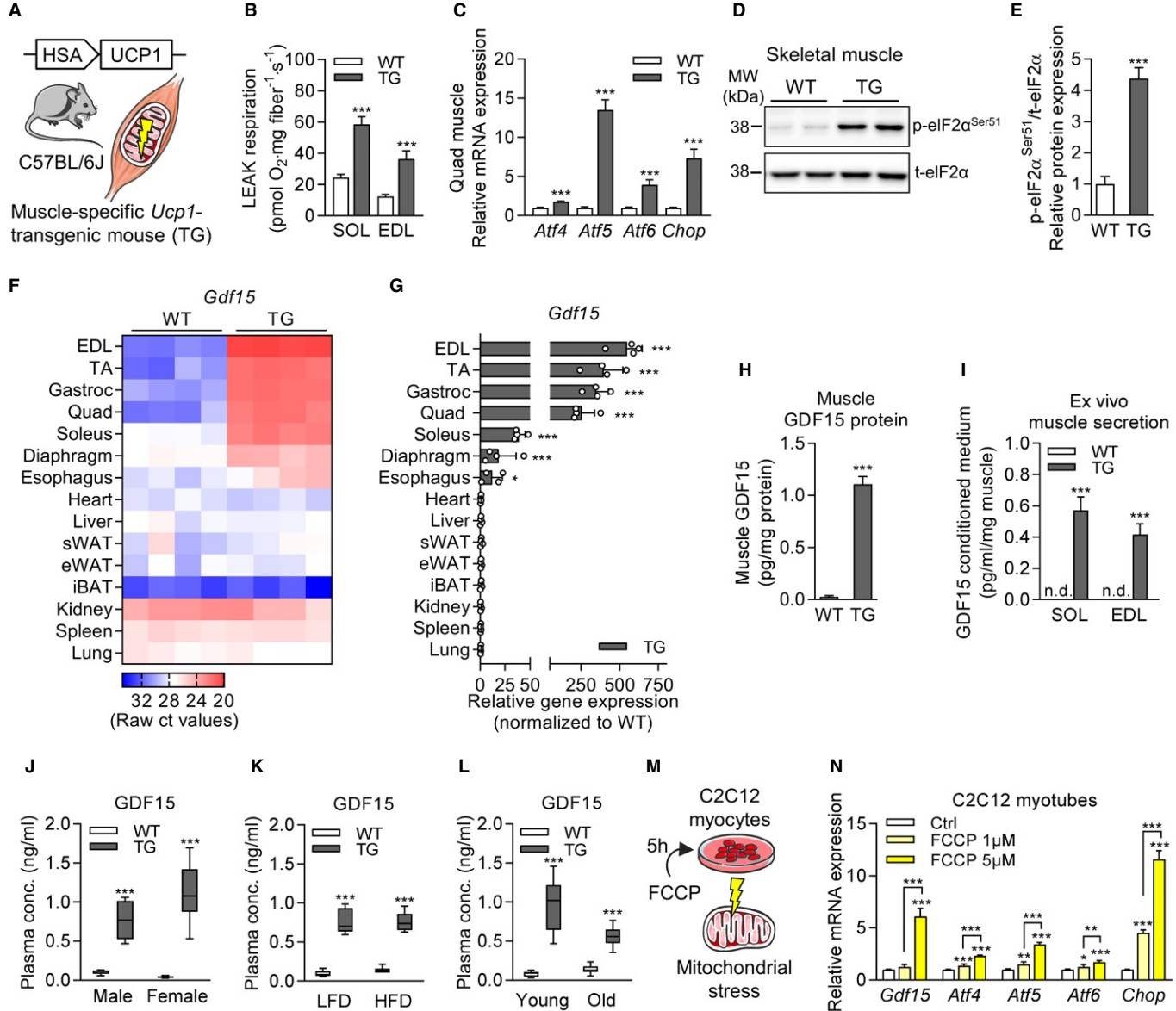

**Figure 1. Muscle mitochondrial stress promotes GDF15 as a myokine.**

A Schematic representation of HSA-*Ucp1*-transgenic (TG) mouse line as model of compromised skeletal muscle-specific mitochondrial OXPHOS capacity via respiratory uncoupling.

B Uncoupled (LEAK) mitochondrial respiration of soleus (SOL) and extensor digitorum longus (EDL) muscle of wild-type (WT) versus *Ucp1*-TG (TG) mice (WT *n* = 9, TG *n* = 5).

C Relative mRNA expression in quadriceps (Quad) of ISR components (*n* = 8 per genotype).

D, E Representative immunoblots of ISR component eIF2α and quantification of phospho-eIF2α (p-eIF2a$^{Ser51}$) relative protein expression in quadriceps skeletal muscle (WT *n* = 3, TG *n* = 4).

F, G Multi-tissue transcriptomic profiling of *Gdf15* gene expression. Heatmap is shown as raw ct expression values (*n* = 4 per genotype). Quantification of *Gdf15* mRNA expression in TG mice is shown as fold change compared to WT littermates (WT *n* = 5, TG *n* = 4).

H Skeletal muscle (Quad and Gastroc) GDF15 protein content normalized to total protein content (WT *n* = 9, TG *n* = 18).

I *Ex vivo* secretion of GDF15 from SOL and EDL muscle of WT versus TG mice (*n* = 6 per genotype) after 2-h incubation normalized to muscle wet weight (mg).

J Mouse GDF15 plasma levels in post-absorptive state of male and female WT versus TG mice at 20 weeks of age (male WT *n* = 8, TG *n* = 9; female WT *n* = 5, TG *n* = 6).

K Mouse GDF15 plasma levels in post-absorptive state of male WT versus TG mice fed low-fat diet (LFD) or high-fat diet (HFD) at 24 weeks of age (LFD WT *n* = 6, TG *n* = 5; HFD WT *n* = 5, TG *n* = 5).

L Mouse GDF15 plasma levels in post-absorptive state of young (10 weeks) versus old (95 weeks) male WT and TG mice (young WT *n* = 10, TG *n* = 13; old WT *n* = 13, TG *n* = 13).

M Representative scheme of *in vitro* study design in differentiated C2C12 myocytes.

N *Gdf15* gene expression of differentiated C2C12 myotubes treated with vehicle control (Ctrl) or chemical mitochondrial uncoupler (FCCP, 1 μM versus 5 μM) for 5 h (*n* = 3 biological replicates).

Data information: Circulating plasma parameters are expressed as interleaved box and whiskers (min to max) plots, and all other data are expressed as means ± SEM; *P*-value calculated by unpaired *t*-test (A–L) or one-way ANOVA with Tukey's post hoc test (N); *$P < 0.05$, **$P < 0.01$, ***$P < 0.001$.

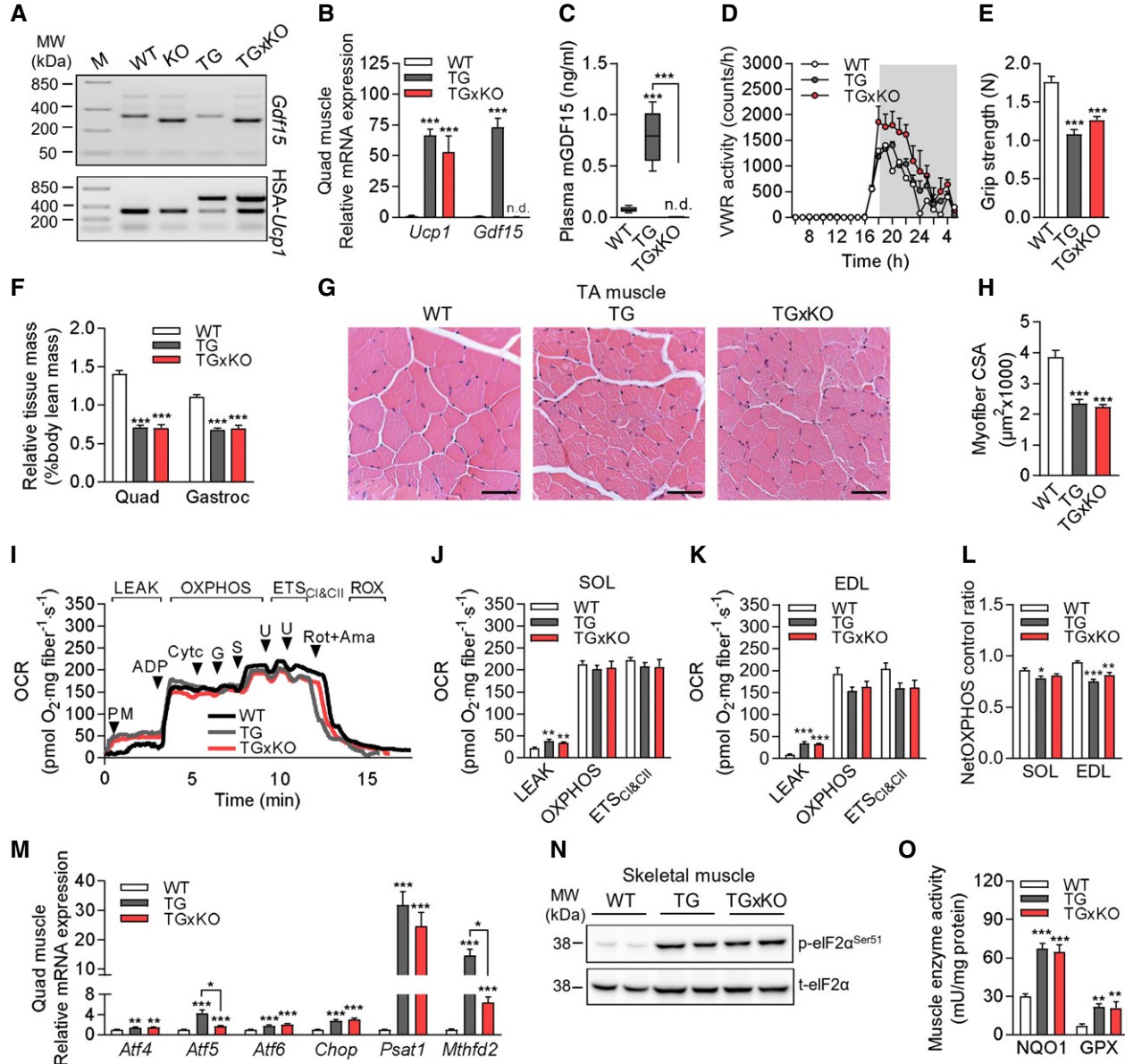

**Figure 2. GDF15-independent muscle wasting and mitochondrial integrated stress response.**

A    Genotyping PCR panel of *Gdf15* and HSA-*Ucp1* loci shown for wild-type (WT), *Gdf15*-KO (KO), *Ucp1*-TG (TG), and *Ucp1*-TGx*Gdf15*-KO (TGxKO) mice.

B    Relative mRNA expression in quadriceps (Quad) of *Ucp1* and *Gdf15* in 20-week-old male WT (*n* = 8), TG (*n* = 7), and TGxKO (*n* = 6) mice.

C    Plasma mGDF15 levels at 20 weeks of age (WT *n* = 9, TG *n* = 10, TGxKO *n* = 9).

D    Voluntary wheel running (VWR) activity shown hourly over 24 h at 15 weeks of age (*n* = 4 per genotype).

E    Grip strength at 10–20 weeks of age (WT *n* = 15, TG *n* = 14, TGxKO *n* = 11).

F    Skeletal muscle mass relative to body lean mass of quadriceps (Quad) and gastrocnemius (Gastroc) from mice at 95 weeks of age (WT *n* = 8, TG *n* = 5, TGxKO *n* = 5).

G, H  Representative H&E histological staining of tibialis anterior (TA) muscle (G) (scale bars represent 50 μm), and cross-sectional area (CSA) of myofibers (H) at 95 weeks (WT *n* = 12, TG *n* = 8, TGxKO *n* = 8).

I    Representative traces of oxygen consumption rate (OCR) during substrate–uncoupler–inhibitor titration (SUIT) protocol for mitochondrial respiratory capacity in permeabilized mouse soleus muscle fibers, PM (pyruvate+malate; LEAK respiration), ADP (OXPHOS capacity), Cyt c (cytochrome c, integrity of outer mt-membrane), G (glutamate;), S (succinate), U (uncoupler, FCCP), Rot (rotenone), and Ama (antimycin A; less than 2% residual oxygen consumption, ROX).

J–L  Mitochondrial respiratory capacity (oxygen consumption rate, OCR) from oxidative soleus (SOL) (J) and glycolytic extensor digitorum longus (EDL) muscle fibers (K), and NetOXPHOS control ratio of 95-week-old mice (L) (*n* = 5 per genotype).

M    Quadriceps (Quad) relative mRNA expression of ISR components at 95 weeks of age (WT *n* = 8, TG *n* = 5, TGxKO *n* = 5).

N    Representative immunoblots of ISR component eIF2α in quadriceps (Quad) muscle of 95-week-old mice.

O    Gastrocnemius enzyme activity of NAD(P)H quinone dehydrogenase 1 (NQO1) and total glutathione peroxidase (GPX) in 95-week-old male mice (WT *n* = 8, TG *n* = 5, TGxKO *n* = 5).

Data information: Circulating plasma parameters are expressed as interleaved box and whiskers (min to max) plots, and all other data are expressed as means ± SEM; *P*-value calculated by one-way ANOVA with Tukey's post hoc test; *$P < 0.05$, **$P < 0.01$, ***$P < 0.001$.

muscles, while only *Atf5* induction was reduced in aged TGxKO mice. Previously, we have shown a ISR-driven remodeling of amino acid and one-carbon metabolism in skeletal muscle of TG mice [7], which was later confirmed in a mouse model for mitochondrial myopathy [8]. The expression level of the enzyme *Psat1*, involved in *de novo* serine synthesis, remained highly elevated, whereas expression of *Mthfd2* as marker of one-carbon metabolism expression was decreased in TGxKO versus TG animals (Fig 2M). Furthermore, chronic mild mitochondrial stress activates the anti-oxidative capacity via NAD(P)H quinone dehydrogenase 1 (NQO1) and total glutathione peroxidase (GPX) induction [37]. We could confirm the induction of NQO1 and GPX in TG muscle, which remained highly induced in TGxKO mice (Fig 2O). Under basal, non-mitochondrial stress conditions, ISR markers as well as activity of anti-oxidant enzymes remained unaffected in KO animals (Fig EV1K–M). Collectively, these findings suggest that GDF15 has neither a protective nor a detrimental cell-autonomous action during muscle mitochondrial stress.

### Genetic ablation of GDF15 drives adiposity during mitochondrial dysfunction

To determine whether muscle mitochondrial stress-induced GDF15 impacts systemic metabolic phenotype, body mass development in early life as well as body composition during aging was monitored. Previous studies showed that the hepatic overexpression of GDF15 in mice lead to decreased body weight and fat mass under a normal chow diet [38,39]. In young adult mice up to 20 weeks of age, the decreased body mass of TG mice was partly recovered in TGxKO animals (Fig 3A). Importantly, this was not due to changes in body lean mass but a substantial increase in body fat mass of TGxKO mice (Fig 3B and C). With progressive aging, TGxKO mice fully recovered their body mass via body fat mass expansion while body lean mass remained unaffected (Fig 3D–F). In line with body fat mass, we found the subcutaneous and epididymal white adipose tissue (sWAT and eWAT) depot weights increased in TGxKO mice compared to TG and WT controls (Fig 3G and H). Male KO mice were indistinguishable in body composition and adipose tissue phenotype from age-matched WT controls (Fig EV3A–H). Thus, genetic ablation of GDF15 during chronic muscle mitochondrial dysfunction promotes a robust fat mass expansion.

### Genetic ablation of GDF15 abolishes white adipose tissue browning and systemic insulin sensitivity during mitochondrial dysfunction

Based on the above results, we tested whether genetic loss of GDF15 during muscle mitochondrial dysfunction further affects systemic metabolic responses. Recently, we and others have demonstrated that mitochondrial dysfunction in muscle promotes metabolic activation of WAT depots [23,40–43]. In line with body composition and adipose tissue mass, male KO mice showed no

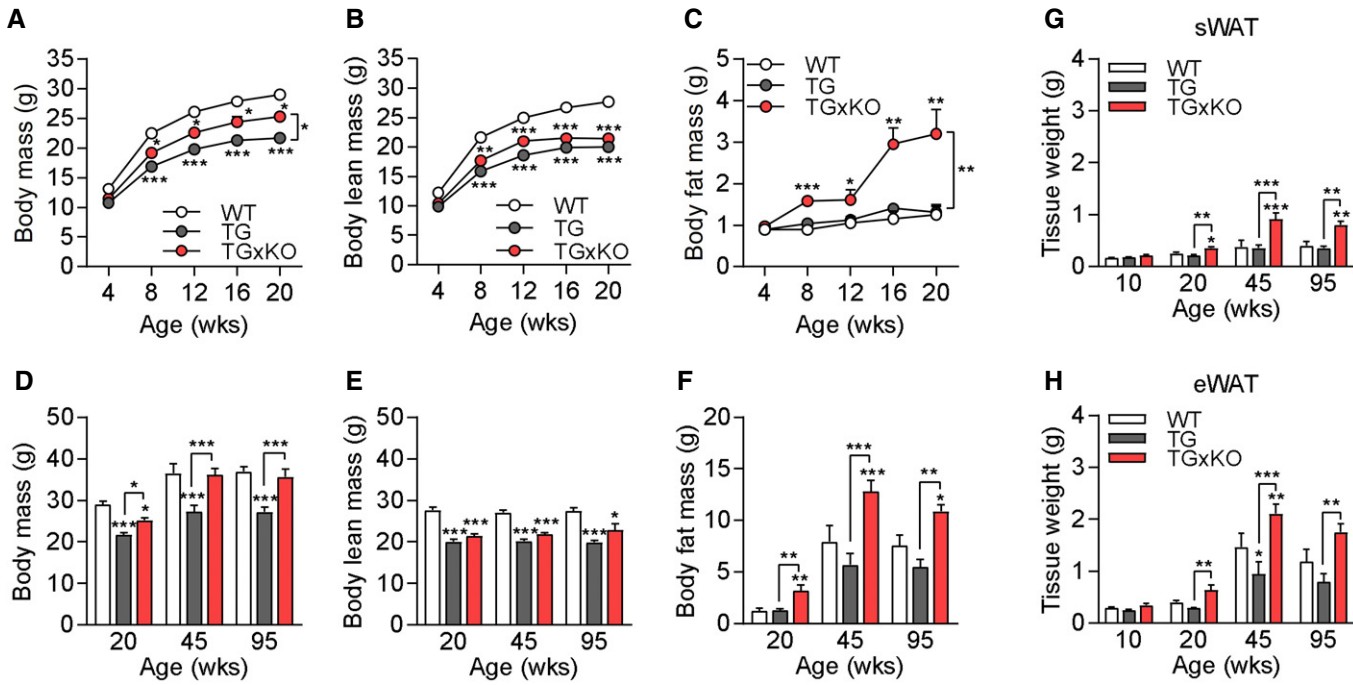

**Figure 3. Genetic ablation of GDF15 drives adiposity during mitochondrial dysfunction.**

A–C   Body mass (A), body lean mass (B), and body fat mass (C) development.
D–F   Body mass (D), body lean mass (E), and body fat mass (F) during aging at 20, 45, and 95 weeks of age.
G, H   Subcutaneous white adipose tissue (sWAT) (G) and epididymal white adipose tissue (eWAT) (H) mass development at 10, 20, 45, and 95 weeks of age.

Data information: Data are from male wild-type (WT), *Ucp1*-TG (TG), and *Ucp1*-TGx*Gdf15*-KO (TGxKO) mice at 10 weeks (WT *n* = 6, TG *n* = 7, TGxKO *n* = 5), 20 weeks (WT *n* = 10, TG *n* = 10, TGxKO *n* = 10), 45 weeks (WT *n* = 5, TG *n* = 6, TGxKO *n* = 7), and 95 weeks (WT *n* = 8, TG *n* = 5, TGxKO *n* = 5). All data are expressed as means ± SEM; *P*-value calculated by one-way ANOVA with Tukey's post hoc test; **P* < 0.05, ***P* < 0.01, ****P* < 0.001.

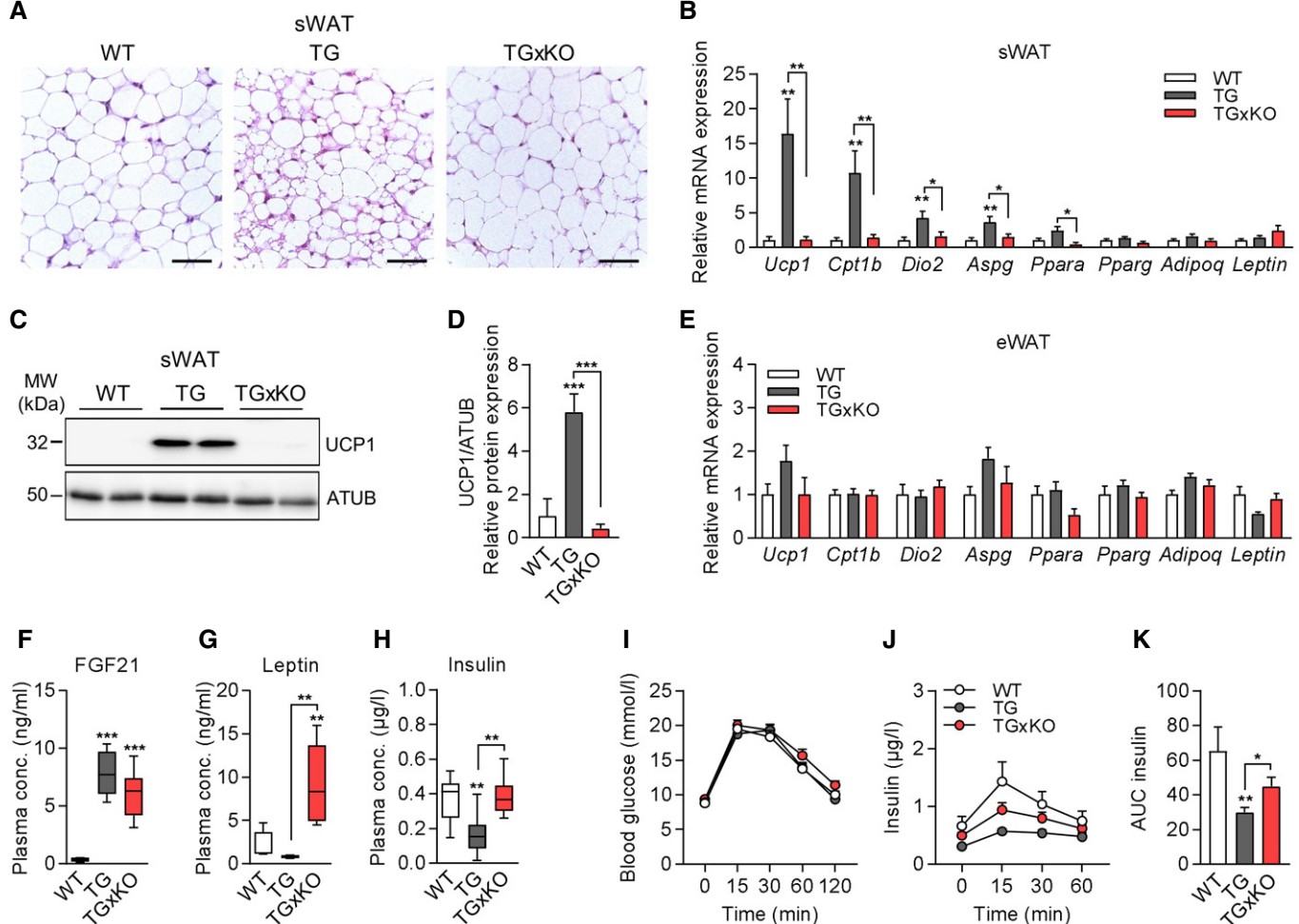

**Figure 4. Genetic ablation of GDF15 abolishes white adipose tissue browning and insulin sensitivity during mitochondrial dysfunction.**

A, B  Representative H&E histological staining of subcutaneous white adipose tissue (sWAT) at 20 weeks of age (scale bars represent 50 μm) (A) and relative mRNA expression profile in sWAT of male mice (WT *n* = 7, TG *n* = 8, TGxKO *n* = 8) (B).

C, D  Representative immunoblots (C) and quantification (D) of UCP1 protein expression in sWAT of male mice at 20 weeks of age (WT *n* = 4, TG *n* = 6, TGxKO *n* = 4).

E  Relative mRNA expression profile in epididymal white adipose tissue (eWAT) (WT *n* = 7, TG *n* = 8, TGxKO *n* = 8).

F  Plasma FGF21 levels from male mice at 20 weeks of age (*n* = 10 per genotype).

G  Plasma Leptin levels from male mice at 20 weeks of age (*n* = 5 per genotype).

H  Post-absorptive plasma insulin levels at 20 weeks of age (WT *n* = 8, TG *n* = 9, TG *n* = 9).

I–K  Blood glucose (I) and insulin levels (J) with total area under the curve (AUC) of insulin (K) during oral glucose tolerance test (oGTT) at 17 weeks of age (WT *n* = 8, TG *n* = 11, TG *n* = 11).

Data information: Data are from male wild-type (WT), *Ucp1*-TG (TG), and *Ucp1*-TGx*Gdf15*-KO (TGxKO) mice. Circulating plasma parameters are expressed as interleaved box and whiskers (min to max) plots, and all other data are expressed as means ± SEM; *P*-value calculated by one-way ANOVA with Tukey's post hoc test; *$P$ < 0.05, **$P$ < 0.01, ***$P$ < 0.001.

differences in sWAT morphology, gene expression of browning markers, or UCP1 protein expression (Fig EV4A–C). Notably, sWAT browning with enhanced multilocular morphology (Fig 4A) and induced gene expression of brown fat markers observed in TG mice was fully blunted in adult TGxKO mice at 20 weeks of age (Fig 4B). In line with the transcriptional profile, UCP1 protein expression was highly induced in sWAT of TG mice, but absent in TGxKO mice (Fig 4C and D). In visceral eWAT, known as the classic WAT depot [44], no mRNA induction of brown fat markers was detected in both TG and TGxKO animals (Fig 4E), as well as KO control mice (Fig 4D). Despite the robust fat mass expansion observed in TGxKO

mice, mRNA expression of *Pparg* as a transcriptional master regulator of adipogenesis as well as of the classic adipokines *Leptin* and *Adipoq* remained unaffected in both sWAT and eWAT depots in TG and TGxKO (Fig 4B and E).

Previously, a link between fibroblast growth factor 21 (FGF21), another well-established marker of mitochondrial diseases, and sWAT browning during mitochondrial dysfunction was proposed [25,40,42,43,45]. Remarkably, the loss of sWAT browning in TGxKO animals occurred despite high circulating FGF21 levels (Fig 4F). Consistent with body fat mass differences, plasma leptin levels were lowest in TG and elevated in TGxKO mice (Fig 4G). Finally,

post-absorptive hypoinsulinemia (Fig 4H) and improved systemic insulin sensitivity of TG mice were abolished in TGxKO mice (Fig 4I–K). Importantly, male KO mice showed no differences in plasma levels of FGF21, leptin, and insulin as well as in insulin sensitivity compared to WT mice (Fig EV4E–J). These data suggest a crucial role of GDF15 as endocrine acting metabolic cytokine in response to muscle-specific mitochondrial dysfunction.

**Mitochondrial stress-induced GDF15 drives daytime-restricted anorectic action**

Next, to understand how mitochondrial stress-induced GDF15 as a myokine regulates fat mass balance and systemic metabolic response we performed a comprehensive *in vivo* metabolic

phenotyping via indirect calorimetry with simultaneous measurement of food intake, energy expenditure, and physical cage activity. Adult mice of all four genotypes at around 17 weeks of age were continuously recorded for 3 days and the last 24 h quantitatively analyzed. Consistent with the above described phenotypic parameters, KO controls showed no differences in any metabolic parameter from age-matched WT mice (Fig EV5A–L).

There were no differences in physical activity (Fig 5A and B) but intriguingly, we observed a shift in the circadian course of food intake and energy expenditure in TG mice which was prevented in TGxKO animals (Fig 5C and E). In TG mice, total energy expenditure (Fig 5D) was reduced in particular during the day, consistent with a daytime-restricted suppression of food intake (Fig 5F and G). Remarkably, genetic ablation of GDF15 fully abolished these effects

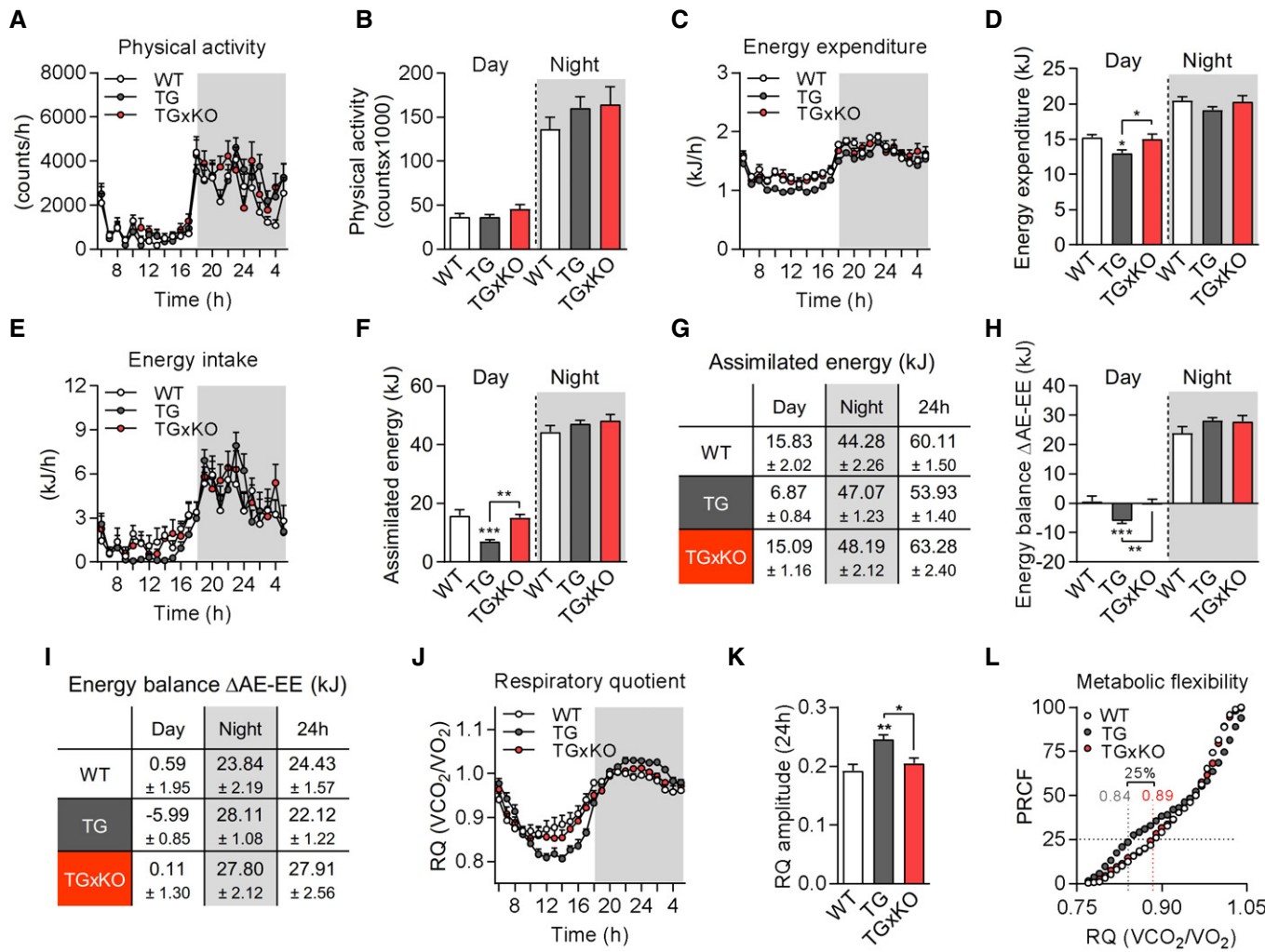

**Figure 5. Muscle-derived GDF15 promotes diurnal anorexia during mitochondrial dysfunction.**

A–F   Physical activity (A, B), energy expenditure (C, D), total assimilated energy (E, F) of male mice shown hourly over 24 h and as day and night time.
G   Table showing mean values of assimilated energy (kJ) at day/night time and per 24 h.
H   Energy balance (kJ) calculated as delta of assimilated energy (AE) and energy expenditure (EE).
I   Table showing mean values of energy balance (ΔAE-EE, kJ) at day/night time and per 24 h.
J–L   Respiratory quotient (RQ) shown hourly over 24 h (J), RQ amplitude (K), and metabolic flexibility via percentage relative cumulative frequency (PRCF) (L).

Data information: Data shown are from male wild-type (WT), *Ucp1*-TG (TG), and *Ucp1*-TGx*Gdf15*-KO (TGxKO) mice at 17–18 weeks of age (WT $n$ = 12, TG $n$ = 11, TG $n$ = 10) and shown as means ± SEM; $P$-value calculated by one-way ANOVA with Tukey's post hoc test; *$P$ < 0.05, **$P$ < 0.01, ***$P$ < 0.001.

in TGxKO animals. Accordingly, overall energy balance was negative during daytime in TG mice (−5.99 kJ ± 0.85) which was abrogated in TGxKO mice (Fig 5H). In contrast, assimilated energy (Fig 5F and G) and energy balance (Fig 5H and I) were higher in both TG and TGxKO mice compared to WT controls during the activity phase at night. Finally, calculation of the 24-h energy balance revealed an excess daily energy incorporation by TGxKO mice of +3.48 kJ and +5.79 kJ compared to WT and TG mice, respectively (Fig 5I). This daily energy surplus is sufficient to explain the differences in body fat mass accumulation. The loss of GDF15-dependent daytime-restricted anorexia together with a

preserved elevated night time energy intake thus drives the progressive fat accumulation observed in TGxKO mice. As a consequence of the diurnal variation of energy balance, skeletal muscle mitochondrial stress in TG mice promotes systemic metabolic flexibility as evident by an increased amplitude (Fig 5J and K) and shift of percent relative cumulative frequency (PRCF) (Fig 5L) of the respiratory quotient (RQ) [46]. Strikingly, this increased metabolic flexibility was fully absent in TGxKO mice. Taken together, we here demonstrate for the first time a GDF15-dependent diurnal anorexic response that reprograms systemic energy homeostasis and metabolic flexibility.

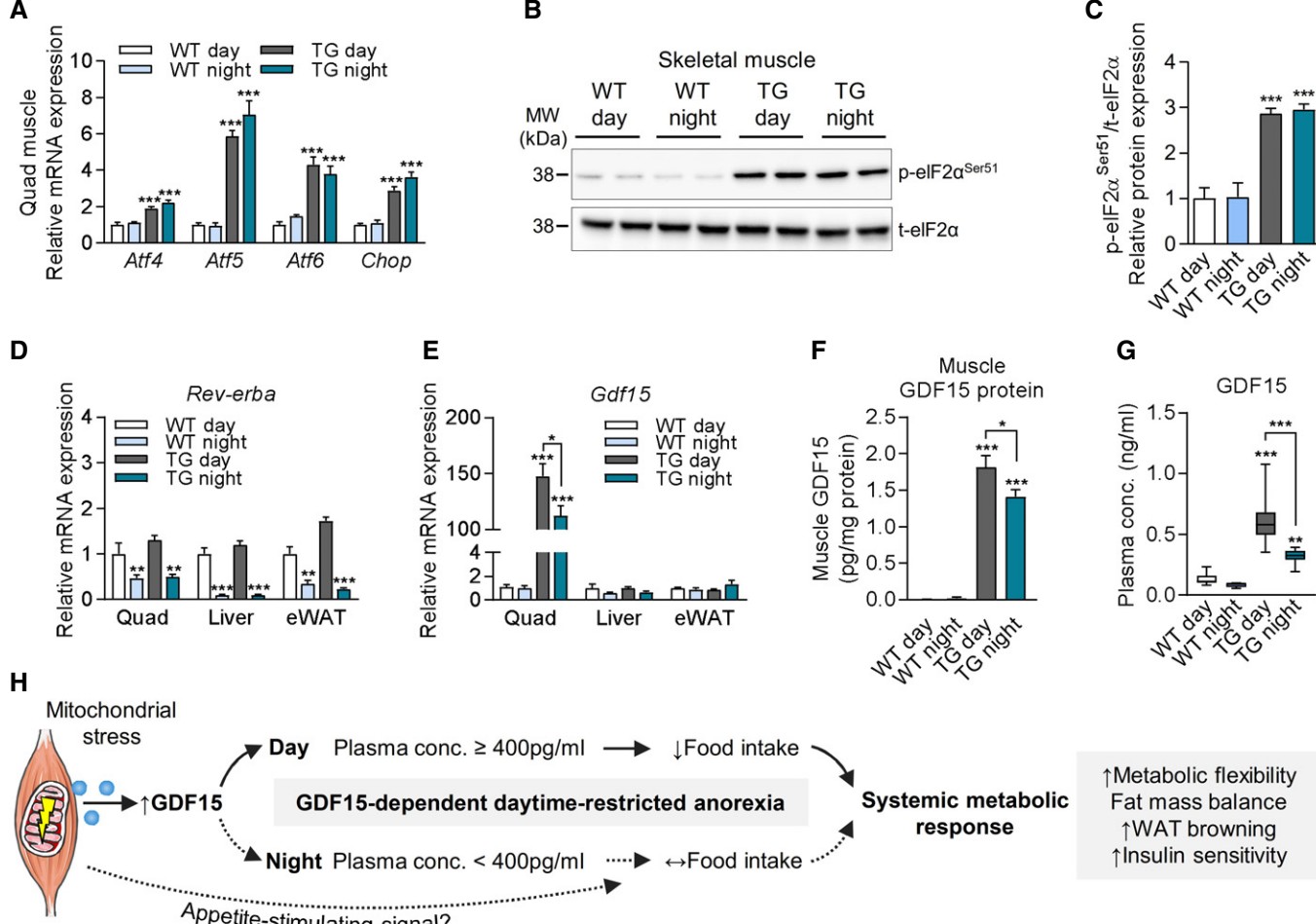

**Figure 6. Diurnal variation of muscle-derived GDF15 during mitochondrial dysfunction.**

A      Relative quadriceps (Quad) mRNA expression of the ISR components *Atf4, Atf5, Atf6,* and *Chop* (WT day $n = 9$, WT night $n = 5$, TG day $n = 20$, TG night $n = 12$).

B, C   Representative immunoblots of ISR component eIF2α (B) and quantification of phospho-eIF2α (p-eIF2aSer51) relative protein expression (C) in skeletal muscle (Gastroc, WT day $n = 4$, WT night $n = 5$, TG day $n = 9$, TG night $n = 9$).

D, E   Relative mRNA expression of *Rev-erba* (D) and *Gdf15* (E) in quadriceps, liver, and eWAT (WT day $n = 9$, WT night $n = 5$, TG day $n = 20$, TG night $n = 12$).

F      Skeletal muscle (Quad) GDF15 protein content normalized to total protein content (WT day $n = 4$, WT night $n = 5$, TG day $n = 14$, TG night $n = 12$)

G      Circulating GDF15 plasma levels (WT day $n = 9$, WT night $n = 5$, TG day $n = 18$, TG night $n = 12$).

H      Graphical summary showing muscle mitochondrial stress induced secretion and diurnal variation of GDF15 to drive a daytime-restricted anorexia, affecting in turn metabolic flexibility, adiposity, WAT browning, and systemic insulin sensitivity. Parts of this cartoon were created using Servier Medical Art (http://smart.servier.com).

Data information: Data shown are from male wild-type (WT) versus *Ucp1*-TG (TG) mice sacrificed at day (10am) or night (10 pm). Circulating plasma parameters are expressed as interleaved box and whiskers (min to max) plots, and all other data are expressed as means ± SEM; *P*-value calculated by one-way ANOVA with Tukey's post hoc test; *$P < 0.05$, **$P < 0.01$, ***$P < 0.001$.

### Diurnal variation of muscle-derived GDF15 during mitochondrial dysfunction

As our data suggest a daytime-restricted action of muscle-derived GDF15, we further investigated whether mitochondrial stress-induced ISR and GDF15 induction differ between day (10 am) and night (10 pm) time. Skeletal muscle gene expression of ISR markers and phosphorylation of eIF2α protein remained highly induced in TG mice at both day and night (Fig 6A–C). We next evaluated the transcriptional regulation of *Rev-erbα*, a well-described regulator of metabolic genes in a circadian manner [47]. Here, we found a regular circadian oscillation of *Rev-erbα* expression in skeletal muscle, liver, and eWAT of both WT and TG mice (Fig 6D), suggesting that the clock molecular machinery was not affected by mitochondrial dysfunction. Interestingly, skeletal muscle-specific induction of *Gdf15* mRNA (Fig 6E) and GDF15 protein expression (Fig 6F) was lower although still significantly elevated in TG mice at night versus day. Circadian *Gdf15* expression remained unaffected in liver and eWAT of TG compared to WT control mice (Fig 6E). Finally, plasma analysis confirmed high circulating GDF15 levels of ≥ 400 pg/ml at day, whereas plasma levels at night dropped consistently to below 400 pg/ml (Fig 5G). The mechanisms behind the diurnal variations in GDF15 plasma and skeletal muscle expression are still to be understood but, altogether, these data suggest that endogenous GDF15 below a certain plasma level is unable to provide an endocrine signal and anorectic action at night during mitochondrial dysfunction. Thus, we propose a diurnal variation and daytime-restricted anorectic action of muscle-derived GDF15 that potentially drives a systemic metabolic response during mitochondrial dysfunction (Fig 6H).

## Discussion

Within the last decade, we and others have proposed a profound importance for inter-tissue communication during mitochondrial stress to control systemic energy homeostasis and metabolic response [23–25,40,42,43,45,48–50]. However, key metabolic mediators and their pathophysiological role in response to mitochondrial dysfunction remained poorly understood. Here, using a transgenic mouse model of mitochondrial stress with compromised muscle-specific OXPHOS capacity via respiratory uncoupling, we (i) demonstrate the induction and diurnal variation of GDF15 as a myokine, (ii) show that genetic ablation of GDF15 during mitochondrial stress prevents a systemic metabolic response, and (iii) provide mechanistic *in vivo* evidence for a GDF15-dependent daytime-restricted anorexia and metabolic flexibility. Hence, we uncovered a novel diurnal action for GDF15 as a myokine in a cell-non-autonomous fashion, whereas this is negligible for mitochondrial stress-driven skeletal muscle atrophy.

A role of endogenous GDF15 as regulator of appetite was originally shown in cancer-related anorexia [26]. Notably, transgenic mice overexpressing GDF15 in the liver have a persistent lean phenotype due to decreased energy intake [38]. Exogenously administrated GDF15 induces an acute anorectic response in mammals including monkeys. This is considered to be centrally mediated by its recently described receptor GFRAL which is exclusively expressed in the hindbrain [27–30]. The anorectic action of GDF15

was proposed to be triggered by neurons within the AP/NTS [51], and ablation of GFRAL completely prevents it. GDF15 and GFRAL are currently discussed as possible targets for the treatment of anorexia/cachexia syndromes [52]. GFRAL knockdown in AP and NTS increased adiposity of mice on a high-fat diet, which shows that the central GDF15-GFRAL axis plays an important role in energy homeostasis [53]. Future investigations may address whether a diurnal anorectic response during muscle mitochondrial dysfunction is based on a muscle–brain axis and requires endogenous GDF15 action in the hindbrain.

In our studies, we demonstrated that the genetic ablation of GDF15 drives adiposity and abolishes sWAT browning and systemic insulin sensitivity as a metabolic response to muscle mitochondrial dysfunction. As pointed out by Breit *et al* [52] there is the possibility that GDF15 effects on other cells or tissues could be mediated by a soluble form of GFRAL resulting from an alternative transcript which lacks transmembrane and cytoplasmatic domains. In contrast, it has been demonstrated previously in several studies that GFRAL is not expressed in WAT depots [27–30,54,55]. However, we cannot rule out a possible direct GDF15-dependent muscle–adipose tissue crosstalk, independent of a GFRAL receptor signaling. Notably, genetic ablation of GDF15 in a mouse model of skeletal muscle-specific deficiency of *Crif1*, an integral protein of the large mitoribosomal subunit, prevented a mitochondrial stress-driven improved insulin sensitivity and resistance to diet-induced obesity [23]. Based on *in vitro* experiments, this was attributed to potential elevated oxidative metabolism and lipid mobilization in the liver, muscle, and WAT depots. However, the direct impact of GDF15 on energy balance, sWAT browning, and systemic insulin sensitivity of muscle-specific *Crif1*-knockout mice under non-dietary challenged conditions was not investigated [23]. We have previously described the myokine FGF21 as an essential regulator of sWAT browning in response to muscle mitochondrial stress [56]. Remarkably, the increase in circulating FGF21 was not affected by loss of GDF15 in TGxKO mice but sWAT browning was still suppressed, which was accompanied by a progressive adiposity. It has been described before that increased adiposity leads to a suppression of sWAT browning [57], as well as a coordinated restructuring of metabolism that could contribute to the whitening of adipose tissue in mice [58]. Thus, we hypothesize that the effects on sWAT browning of high circulating FGF21 are affected by the lack of GDF15, possibly due to an indirect effect via fat mass expansion. Notably, in line with previous studies in obese mice and humans who display a leptin resistance [57,59,60], adiposity in TGxKO mice also promoted an increase in circulating leptin levels suggesting no particular disturbance of the leptin axis in the TG mouse model.

Interestingly, our here uncovered GDF15-dependent diurnal anorexia resembles an endogenous mimic of time-restricted feeding response. In mice and humans, time-restricted feeding without reducing caloric intake improves systemic glucose tolerance, insulin sensitivity, and metabolic flexibility [61–63]. Metabolic flexibility describes efficient switches in metabolism depending on organismal energy demand [64,65], which was previously reported in response to chronic mitochondrial stress in mice [66,67]. Here, we demonstrate that the genetic ablation of GDF15 in TG mice fully abolishes the metabolic flexibility as well as systemic insulin sensitivity. Thus, our findings suggest a crucial function of endogenous GDF15 as an

endocrine acting regulator of systemic metabolic homeostasis and fat mass balance during mitochondrial dysfunction.

In *Drosophila*, a myokine-mediated diurnal inter-tissue communication between muscle and adipose tissue was recently described, affecting systemic energy homeostasis [68]. In healthy humans, it was shown that circulating GDF15 levels vary in a diurnal pattern [69]. Here, we found that despite a preserved skeletal muscle ISR induction, GDF15 expression and circulating plasma concentrations at night were significantly lower than daytime levels. The reasons for this are unclear, but may reflect variations of intracellular processing of GDF15 [70] in a circadian manner. Paradoxically, induction of GDF15 as a myokine was unable to suppress food intake at night, which was rather increased compared to WT control. Whether the anorectic action requires a certain threshold of circulating GDF15 ($\geq$ 400 pg/ml) or an additional appetite-stimulating signal during muscle mitochondrial dysfunction limits GDF15 action at night remains to be elucidated.

Based on our results, we suggest that the GDF15-dependent diurnal anorexia represents the so far missing link between muscle mitochondrial dysfunction and remodeling of systemic energy homeostasis. Circulating GDF15 levels are highly elevated in cancer patients with severe anorexia [26,71,72] and in patients with mitochondrial diseases [19–22]. Noteworthy, poor feeding in infants and multiple episodes of nausea as well as vomiting are common phenotypes in human mitochondrial disorders [73,74]. Interestingly, GDF15 was recently associated with severe nausea, vomiting, weight loss, and anorexia symptoms during pregnancy, known as hyperemesis gravidarum [75]. This raises the intriguing question whether endogenous GDF15 mediates an anorectic response in patients with mitochondrial dysfunction.

In conclusion, our novel findings align with the notion that mitochondrial stress drives an endocrine crosstalk, although further studies are necessary to better understand the central actions of muscle-derived GDF15 during mitochondrial dysfunction. Our evidence contributes to a better understanding of the mechanisms that promote the pathophysiological consequences of mitochondrial dysfunction. Consequently, this should stimulate translational research on GDF15 as a therapeutic target for the treatment of anorexia and hypermetabolism during mitochondrial disorders.

# Materials and Methods

### Animals

Animal experiments were approved by the ethics committee of the Ministry of Agriculture and Environment (State Brandenburg, Germany, permission number GZ 2347-9-2016). HSA-*Ucp1*-transgenic mice (TG) [32] were crossed with whole-body *Gdf15*-knockout (KO) mice [34] kindly provided by Dr. Se-Jin Lee (University of Connecticut School of Medicine, Department of Genetics and Genome Sciences) to generate the four experimental genotypes: wild-type (WT), *Gdf15*-KO (KO), TG, and TGx*Gdf15*-KO (TGxKO) mice. Notably, the human skeletal actin (HSA) promoter of TG mice is well described to drive a skeletal muscle-specific expression [76], in all muscle fiber types but preferentially in fast-glycolytic type muscle fibers [77]. All animals were group-housed and random-caged with *ad libitum* access to a standard chow diet

(Sniff, Soest, Germany) and water at 23°C and a 12:12-h dark-light cycle and kept to different ages. For the experiments, female and male animals were used and sacrificed in the end of the experiment after 3-h food withdrawal to collect plasma and tissue samples.

### In vivo phenotyping

For body composition measurement, quantitative magnetic resonance (QMR, EchoMRI 2012 Body Composition Analyzer, Houston, USA) was used. Voluntary running wheel activity was determined by IR motion detectors (TSE Systems GmbH, Homburg, Germany). Grip strength was measured using a Grip Strength Meter (BIOSEB). Energy expenditure, food intake, physical activity, and respiratory quotient (RQ = $CO_2$ produced/$O_2$ consumed) were assessed by indirect calorimetry using an open respiratory system with the simultaneous measurement of cage activity, food, and water intake (TSE PhenoMaster, TSE Systems, Germany). Measurements were performed in male mice at 17 weeks of age in 10 min intervals over a period of 72 h. The last 24 h were analyzed to assess genotypic differences of energy metabolism. For the oral glucose tolerance test, 2 mg glucose per gram body weight was applied 2 h after food withdrawal. Blood glucose levels were measured before and 15, 30, 60, and 120 min after glucose application using a glucose sensor (Bayer, Germany). Insulin levels were measured before and 15, 30, and 60 min after application by an ultra-sensitive ELISA assay (DRG Instruments GmbH, Germany).

### Ex vivo skeletal muscle culture and cell culture

Skeletal muscle *ex vivo* culture was adapted to a protocol described previously [78]. Briefly, mice were fasted for 4 h prior the dissection of muscles. Mice were sacrificed under isoflurane anesthesia. Extensor digitorum longus (EDL) and soleus muscles were rapidly removed, washed, and incubated in pre-oxygenated (95% $O_2$/5% $CO_2$) Krebs–Henseleit buffer (KHB) supplemented with 15 mM mannitol and 5 mM glucose for 2 h at 30°C in a humidified incubator containing 5% $CO_2$. Conditioned medium was centrifuged, and supernatant was stored at $-20$°C until analysis. C2C12 cells were cultured in DMEM medium supplemented with 10% FBS and 1% penicillin-streptomycin at 37°C in a 5% $CO_2$ atmosphere. When C2C12 cells reached to 90% confluence, differentiation was induced by incubation for 6 days with DMEM medium containing 2% horse serum and 1% penicillin-streptomycin. For RNA isolation, the differentiated C2C12 muscle cells were treated with vehicle control (Ctrl) or mitochondrial uncoupler (FCCP, 5 $\mu$M), for 5 h ($n = 3$ biological replicates).

### Gene expression analysis

RNA isolation and quantitative real-time PCR from cells and tissues were performed as described previously [79]. Briefly, total RNA was isolated using peqGOLD TriFast (#732-3314) from VWR, followed by a DNase digest treatment (#EN0521, Fischer Scientific). cDNA synthesis was performed using the RevertAid First Strand cDNA Synthesis Kit (#K1622) from Thermo Fischer Scientific from 1 $\mu$g of RNA. Quantitative real-time PCR analyses were performed on a ViiA™ 7 Real-Time PCR System from Applied Biosystems using

384-well plates (#72.1984.202, Sarstedt). The qPCR mix contained 5 ng of cDNA, Power SYBR Green Master Mix (#4367660, Thermo Fischer Scientific), and 1.5 µM of each corresponding primer pair in a total volume of 5 µl per well. The following primers were used: *Actb*: 5′ GCCAACCGTGAAAAGAGAC 3′ (F), 5′ TACGACCAGAGG CATACAG 3′ (R); *Atf4*: 5′ GGAATGGCCGGCTATGG 3′ (F), 5′ TC CCGGAAAAGGCATCCT 3′ (R); *Atf5*: 5′CTACCCCTCCATTCCACTT TCC 3′ (F), 5′TTCTTGACTGGCTTCTCACTTGTG 3′ (R); *Atf6*: 5′ CTTCCTCCAGTTGCTCCATC 3′ (F), 5′ CAACTCCTCAGGAACGTGC T 3′ (R); *B2 m:* 5′ CCCCACTGAGACTGATACATACGC 3′ (F), 5′ AGAAACTGGATTTGTAATTAAGCAGGTTC3′ (R); *Chop*: 5′ AGAGT GGTCAGTGCGCAGC 3′ (F), 5′ CTCATTCTCCTGCTCCTTCTCC 3′ (R); *Gdf15*: 5′ GAGCTACGGGGTCGCTTC 3′ (F), 5′ GGGACCC CAATCTCACCT 3′ (R); *Psat1*: 5′ AGTGGAGCGCCAGAATAGAA 3′ (F), 5′ CTTCGGTTGTGACAGCGTTA 3′ (R); *Mthfd2*: 5′ CATGGGG CGTGTGGGAGATAAT 3′ (F), 5′ CCGGGCCGTTCGTGAGC 3′ (R); *Ucp1*: 5′ TGGAGGTGTGGCAGTATTC 3′ (F), 5′ AGCTCTGTACAGTT GATGATGAC 3′ (R); *Cpt1b*: 5′ GAAGAGATCAAGCCGGTCAT 3′ (F), 5′ CTCCATCTGGTAGGAGCACA 3′ (R), 6-FAM-TGGGCACCATACC CAGTGCC-TAMRA (P), for which TaqMan™ Gene Expression Master Mix (#10015414, Fischer Scientific) was used; *Cidea*: 5′ TGCTCTTCTGTATCGCCCAGT 3′ (F), 5′ GCCGTGTTAAGGAATC TGCTG 3′ (R); *Dio2*: 5′ TGCCACCTTCTTGACTTTGC 3′ (F), 5′ GGTTCCGGTGCTTCTTAACC 3′ (R); *Elovl3*: 5′ TCCGCGTTCTCATG TAGGTCT 3′ (F), 5′ GGACCTGATGCAACCCTATGA 3′ (R); *Fgf21*: 5′ GCTGCTGGAGGACGGTTACA 3′ (F), 5′ CACAGGTCCCCAG GATGTTG 3′ (R); *Aspg*: 5′ AGGCATCAGAGTGTCATT 3′ (F), 5′ GGCACAGTGTCCATCATA 3′ (R); *Pgc1a*: 5′ AGCCGTGACCACT GACAACGAG 3′ (F), 5′ GCTGCATGGTTCTGAGTGCTAAG 3′ (R); *Prdm16*: 5′ CAGCACGGTGAAGCCATTC 3′ (F), 5′ GCGTGCAT CCGCTTGTG 3′ (R). *Rev-erba*: 5′ CGTTCGCATCAATCGCAACC 3′ (F), 5′ GATGTGGAGTAGGTGAGGT 3′ (R). Relative gene expression to WT controls was calculated as ΔCT using *B2 m* or *Actb* for normalization.

## Plasma and cytokine analysis

Blood glucose concentrations were determined in samples from lateral tail vein using a glucose sensor (Bayer, Germany). Post-absorptive plasma insulin levels were measured by an ultra-sensitive ELISA assay (DRG Instruments GmbH, Germany). The plasma concentrations of cytokines (Mouse/Rat GDF15, FGF21, and Leptin Quantikine ELISA Kit, R&D Systems, USA) were determined by enzyme-linked immunosorbent assays (ELISA) following manufacturer instructions. Mouse GDF15 (mGDF15) protein levels were analyzed in skeletal muscle lysates of quadriceps from 20-week-old male WT and *Ucp1*-TG animals. Muscle tissue was homogenized in hypotonic buffer (10 mM Tris pH 7.4, 0.02% Triton X-100, 1× COMPLETE EDTA-free), GDF15 was measured using Mouse/Rat GDF15 ELISA Kit (R&D Systems), and values were normalized to total protein concentration (*DC*™ Protein Assay Reagent [#500-0114, BioRad]).

## Immunoblotting

For protein extraction, tissue was homogenized in RIPA buffer containing protease and phosphatase inhibitor cocktail (#A32959, Thermo Scientific) and protein concentration was measured using the *DC*™ Protein Assay Reagent (#500-0114, BioRad). The following primary antibodies were used: phospho-AMPK (Thr172) (#2531, Cell Signaling Technology), total AMPK (#2603, Cell Signaling Technology), phospho-eIF2α (Ser51) (#3597, Cell Signaling Technology), eIF2α antibodies (#3524, Cell Signaling Technology), OXPHOS antibody (#ab110413, Abcam), slow myosin (#ab11083, Abcam), and UCP1 antibodies (#ab23841, Abcam). Protein expression was normalized to α-Tubulin (ATUB) (#T6074, SIGMA). Horseradish peroxidase-conjugated secondary antibodies were used: anti-rabbit IgG (#7074, Cell Signaling Technology) or anti-mouse IgG (#7076, Cell Signaling Technology).

## Histology

Muscle (tibialis anterior [TA]), and subcutaneous white adipose tissue (sWAT) pieces were fixed in 4% formaldehyde for 24 h and embedded in paraffin. Tissue slices were cut (2–4 µM, Microm HM 355 S Rotatory Microtome), mounted, and dehydrated in increasing ethanol series for hematoxylin and eosin (H&E) (Roth, Fluka) staining. Bright field images were taken with an Axioplan 2 (Zeiss) using an AxioCam MRm camera. Quantification of cross-sectional area was performed using ImageJ (version 1.52k).

## OROBOROS high-resolution respirometry

Mitochondrial respiration *ex vivo* analysis was performed in soleus and extensor digitorum longus (EDL) muscle as previously described using the high-resolution Oxygraph-2k (OROBOROS Instruments, Innsbruck, Austria) [37]. Briefly, muscle samples were gently dissected, immediately placed in ice-cold biopsy preservation medium (BIOPS: 2.77 mM CaK$_2$EGTA, 7.23 mM K$_2$EGTA, 5.77 mM Na$_2$ATP, 6.56 mM MgCl$_2$•6 H$_2$O, 20 mM taurine, 15 mM Na$_2$phosphocreatine, 20 mM imidazole, 0.5 mM dithiothreitol, 50 mM MES hydrate, pH 7.1) [80], separated with a pair of fine-tipped forceps, and finally permeabilized with saponin (50 µg/ml) for 30 min at 4°C. After permeabilization, muscle fibers were washed in mitochondrial respiration medium (Mir05: 0,5 mM EGTA, 3 mM MgCl$_2$•6H$_2$O, 60 mM K-lactobionate, 20 mM taurine, 10 mM KH$_2$PO$_4$, 20 mM HEPES, 110 mM sucrose, 1 g/l fatty acid-free BSA) for 10 min at 4°C and kept on ice until analysis. Respiratory capacity was analyzed performing a multiple substrate–uncoupler–inhibitor titration (SUIT) protocol [81,82] at 37°C in a hyper-oxygenated environment using following substrate concentrations: 0.5 mM malate + 5 mM pyruvate (LEAK respiration [L]), 5 mM ADP, 10 µM cytochrome c (integrity of outer mt-membrane, data not shown), 10 mM glutamate + 10 mM succinate (OXPHOS [P]), 0.5 µM FCCP (ETS$_{CI\&CII}$, electron transfer system, [E]), and 0.5 µM rotenone + 2.5 µM antimycin A (residual oxygen consumption, ROX, data not shown). Oxygen flux was quantified using DatLab software (version 6, OROBOROS Instruments, Innsbruck, Austria). Oxygen consumption rate (OCR) was normalized to muscle wet weight of dry blotted fiber bundles. The netOXPHOS control ratio ([P-L]/E) was calculated to express the OXPHOS capacity (corrected for LEAK respiration) as a fraction of ETS capacity.

## Statistical analysis

Statistical analyses were performed using Stat GraphPad Prism 7 (GraphPad Software, San Diego, CA, USA). All plasma components

are expressed as interleaved box and whiskers (min to max) plots, and all other data are expressed as means with SEM. After testing for equal distribution of the data and equal variances with the data-sets using D'Agostino & Pearson normality test and Student's *t*-test (unpaired, 2-tailed), one-way ANOVA followed by the Tukey's multiple comparison test was used to determine differences between genotypes, treatments, or time points. Statistical significance was assumed at $P < 0.05$ and denoted by $*P < 0.05$, $**P < 0.01$, $***P < 0.001$. Means with asterisk are significantly different to WT or indicated group.

**Expanded View** for this article is available online.

## Acknowledgements

This work was supported by the German Research Foundation (DFG, grant KL613/23–1 to SKl) and by COST Action CA15203 MitoEAGLE, supported by COST (European Cooperation in Science and Technology). The authors would like to thank Antje Sylvester, Petra Albrecht, Sandra Rumberger, Carola Gehrmann, Elisabeth Meyer, Dr. Wenke Jonas, and Andrea Teichmann for excellent technical assistance. Parts of graphical abstract as well as Figs 1A and M, and 6H were created using Sevier Medical Art (https://smart.servier.com).

## Author contributions

MO and SKl designed the study. MO and CIG conceived experiments and analyzed the majority of the data. MO, CIG and VC performed the majority of experiments. VV, SE and MW performed transcriptomic and proteomic analyses and analyzed data. SKe performed *ex vivo* muscle secretion assay and helped analyze data. MO and CIG prepared the figures, drafted, and wrote the manuscript, which was substantially proof-read, commented, and edited by all authors.

## Conflict of interest

The authors declare that they have no conflict of interest.

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
