## [Review Process File · EMBO Reports]

Muscle derived GDF15 drives diurnal anorexia and systemic metabolic remodeling during mitochondrial stress

Mario Ost, Carla Igual Gil, Verena Coleman, Susanne Keipert, Sotirios Efstathiou, Veronika Vidic, Miriam Weyers, and Susanne Klaus

Review timeline:

Submission date:	4 July 2019
Editorial Decision:	1 August 2019
Revision received:	30 October 2019
Editorial Decision:	9 December 2019
Revision received:	16 December 2019
Accepted:	10 January 2020

Editor: Deniz Senyilmaz-Tiebe

Transaction Report:

1st Editorial Decision

1 August 2019

Thank you for the submission of your research manuscript to our journal. We have now received the full set of referee reports that is copied below.

As you can see, the referees find the proposed role of GDF15 in relaying the systemic effects of muscle specific mitochondrial dysfunction of interest, but they also raise significant concerns that need to be addressed before considering publication here.

To name a few (not all), referees #1 and 2 find that more conclusive evidence supporting the role of GDF15 in regulation of diurnal food intake is required. Referee #1 would like to see whether chronic GDF15 treatment is sufficient to elicit the effects on diurnal food intake. Referee #3 finds that the source of GDF15 secretion in response to muscle specific mitochondrial stress remains currently elusive. To demonstrate this, muscle specific GDF15 KO mice would need to be employed. I don't know if you already have this strain at hand (if you do, this experiment would strengthen the manuscript), but I have discussed this point with referee #3 and if you cannot address this point, it will not preclude publication here. However, the title will need to be altered accordingly.

Given these constructive comments, we would like to invite you to revise your manuscript with the understanding that the referee concerns (as detailed above and in their reports) must be fully addressed and their suggestions taken on board. Please address all referee concerns in a complete point-by-point response. Acceptance of the manuscript will depend on a positive outcome of a second round of review. It is EMBO reports policy to allow a single round of revision only and acceptance or rejection of the manuscript will therefore depend on the completeness of your responses included in the next, final version of the manuscript.

REFeree REPORTS

Referee #1:

In this manuscript the authors have studied the impact of mitochondrial muscle dysfunction induced by overexpression of UCP1 in skeletal muscle on the muscle production of GDF15 and its functional consequences. The authors document that muscle overexpression of UCP1 causes mitochondrial stress and enhances muscle expression of GDF15 and also greater circulating concentrations. UCP1 transgenic mice also showed a reduced food intake during the light phase but not at dark suggesting that perhaps GDF15 regulates the circadian pattern of food intake. The topic is of high interest and the manuscript has essentially two major flaws, namely, the lack of some essential control groups in some of the figures, and the lack of any evidence that GDF15 is the direct regulation of diurnal food intake.

Major comments.

1. The authors define in the title "pseudo-intermittent fasting" to the fact that UCP1 transgenic mice have a lower food intake during the light phase of the day but not during the dark phase. It is unclear to me whether this is precise enough.
2. Figures 2, 3 and 4 miss to include outstanding data from the GDF15 KO mouse, and this is essential to validate some of the major conclusions of the manuscript. Only Figure 4 is complemented by some of those data included as Figure S1. Based on this, it is unclear at this moment, whether ablation of GDF15 is important or not for the effects of UCP1 expression on mitochondrial stress, muscle metabolism, adiposity, or browning of subcutaneous adipose tissue.
3. The authors also document that UCP1 transgenic mice show an alteration in diurnal regulation of food intake. However, they at least should determine whether GDF15 is sufficient to induce these effects in mice. Does chronic GDF15 treatment to normal mice reduce in diurnal food intake?

Referee #2:

The manuscript by Ost and colleagues showed that a transgenic mouse with skeletal muscle-specific mitochondrial OXPHOS defect (Ucp1-TG), had induction of diurnal variation of GDF15. Knockout of GDF-15 in the Ucp1-TG promoted a progressive increase of body fat mass, which was associated with suppression of day time food intake resembling a pseudo-intermittent fasting. This led to adipose tissue remodeling, hypoinsulinemia, and increased energy metabolism. They conclude that GDF15 has an important role in diurnal regulation of energy balance during mitochondrial dysfunction. They also showed that sustained GDF15 elevation is not protective for muscle.

The study is of high quality and the use of genetically modified mice very appropriate to address the scientific question. The manuscript has two main messages. The first one is quite convincing, showing adipose tissue metabolic remodeling blunted in adult TGxKO mice. However, the second one, related to circadian changes, the data was not as strong. I could not really see the elevated carbohydrate oxidation during day time and increased lipid oxidation at night in TG mice, abolished in TGxKO mice (Fig 4C and D). In fact, a lot of the diurnal differences were quite minimal, and one wonders how physiologically relevant they are. It is possible that I am not interpreting the data correctly, so I would be glad to let the authors argue their case.

Referee #3:

In this manuscript by Ost et al., the authors provide evidence that GDF15 is produced in muscle under conditions of chronic mitochondrial uncoupling (overexpression of UCP1) that has mostly peripheral effects (i.e. not in muscle), leading them to conclude muscle-derived GDF15 is a mitokine. Arguably, the most interesting of these peripheral effects is the demonstration of GDF15-dependent suppression of day time food intake (i.e. circadian effects). The study is quite well executed and the primary data are robust. However, there are some concerns with experimental design and other issues that undercut the very strong conclusion reached that need to be addressed. These are detailed below:

1. The overall study design precludes an unequivocal conclusion about muscle being the primary source of the active Gdf15. That is, even though the mitochondrial stress is in muscle due to the

muscle-specific HSA driver (although expression in esophagus and diaphragm should be discussed, McCarthy et al. 2012), the GDF15 KO is global. Thus, the possibility exists that muscle mitochondrial stress is triggering GDF15 expression/secretion elsewhere. Obviously, in this regard, knocking out GDF15 only in muscle is the key experiment. Furthermore, it is also possible that loss of Gdf15 expression in other tissues contributes to the phenotypes observed in the KO and TGxKO mice. The eWAT is a highly responsive fat depot (to diet, hormonal regulation and other stressors). Thus, it would be interesting to see if Gdf15 is expressed here, and if it is released from the eWAT.

2. Although the data presented suggest that there is a minimal effect of increased Gdf15 muscle expression on the muscle itself, it is a stretch for the authors to conclude that there is no direct muscle-adipose tissue crosstalk (line 160-161). For example, UCP1 protein expression in sWAT was apparently completely decreased in TGxKO mice (Fig. 3D) and the mRNA levels of several markers of "browning" were decreased in sWAT of TGxKO. Thus, how can the authors conclude that there was no impact of Gdf15 loss on sWAT? Additional measurements are needed in this regard: for example, quantification of regulated circulating factors that remodel adipose and could contribute to the "browning," effects of GDF15 on eWAT depots, and measurement of WAT differentiation markers expression like PPAR γ .

3. Given the extensive literature demonstrating that Gdf15 regulates adiposity via central regulation of food intake, the authors should measure satiety hormones & peptides (e.g. CART, alpha-MSH, leptin) in the different mouse models and investigate whether Gdf15 impacts their regulation in a circadian manner, and if loss of Gdf15 alters levels or circadian expression of these factors.

4. The authors demonstrate that there is no significant difference in energy intake (Fig. 4H) or energy expenditure (Fig. 4K) between TG-KO and WT mice. However, there appears to be a consistently higher rate of fat mass accumulation in the TG-KO mice relative to the WT mice (Fig. 3A-C). What was the statistical significance for differences between TG-KO and WT for the fat mass data? How do the authors reconcile the lack of any apparent difference in food intake between WT and TG-KO with the relatively higher fat mass accumulation in the TG-KO mice? The authors state in line 198-199 that the increased daytime energy intake "can completely explain the progressive fat accumulation observed in TGxKO mice." This is a very aggressive statement and the data presented in the manuscript do not provide enough evidence to make this conclusion. The authors should consider subjecting the TGxKO, WT and TG mice to a pair-feeding regimen to determine whether equalizing energy intake across all groups will reverse the differences in fat mass accumulation.

5. The transcriptional regulation of Gdf15 expression by mitochondrial dysfunction, or specifically uncoupling in the UCP1-TG mice, could have been investigated in greater detail. How does mitochondrial uncoupling lead to up-regulation of Gdf15 expression? What are the pathways involved, and what transcriptional regulators are recruited to increase activity at the Gdf15 promoter? The authors suggest (lines 90-91) that a CHOP-dependent mechanism of Gdf15 upregulation might explain the increased Gdf15 observed with mitochondrial uncoupling in the TG mice. However, in Fig. 5 the data clearly suggests that even though expression of CHOP and other ISR mediators is maintained day or night, Gdf15 expression and plasma levels fluctuate significantly. How do the authors explain this?

6. Does the circadian rhythmicity of Gdf15 expression alter as the animal ages? It would be interesting to see if there is a difference in muscle expression, plasma levels, and in the delta change between day/night, as the animal ages.

7. Does HFD feeding impact the circadian rhythmicity of Gdf15 expression? Are there diurnal variations in Gdf15 expression in other tissues?

8. The Introduction and Discussion could be written more succinctly and clearly.

Minor points

- Line 34: "manifest at any age and organ ..."; meaning is unclear
- Line 39: "thereby described firstly ..."; awkward
- Line 50: "secreted molecules" vague, please use mitokine/cytokine/hormone

- Line 70: "activating adipose tissue" vague; what is meant by activating?
- Use of acronyms (especially the name of the mouse Ucp1-TG): once an acronym has been introduced and the full written form has been indicated, the acronym only should be used for all subsequent mentions (no need to repeatedly use the full form).
- The acronym mtISR is introduced at line 91 as a specific type of ISR, but later mentions of ISR are not specified at mtISR (eg. line 136, 208). Why did the authors specify mtISR, and is it necessary? The authors should decide how to reference either ISR or mtISR through the text and evaluate the need for two acronyms.
- Line 204: Were the times for day and night correctly stated? As written in the manuscript, the day phase and night phase overlap.

1st Revision - authors' response

30 October 2019

Referee comments

Referee #1:

In this manuscript the authors have studied the impact of mitochondrial muscle dysfunction induced by overexpression of UCP1 in skeletal muscle on the muscle production of GDF15 and its functional consequences. The authors document that muscle overexpression of UCP1 causes mitochondrial stress and enhances muscle expression of GDF15 and also greater circulating concentrations. UCP1 transgenic mice also showed a reduced food intake during the light phase but not at dark suggesting that perhaps GDF15 regulates the circadian pattern of food intake. The topic is of high interest and the manuscript has essentially two major flaws, namely, the lack of some essential control groups in some of the figures, and the lack of any evidence that GDF15 is the direct regulation of diurnal food intake.

Response:

Thank you for your interest in our work and the constructive remarks and suggestions. We addressed all your concerns as outlined below and hope that you find our revised manuscript acceptable for publication.

Major comments

1. The authors define in the title "pseudo-intermittent fasting" to the fact that UCP1 transgenic mice have a lower food intake during the light phase of the day but not during the dark phase. It is unclear to me whether this is precise enough.

Response: We acknowledge that the term "pseudo-intermittent fasting" in the title might be confusing. We have changed the manuscript title to: "GDF15 drives daytime-restricted anorexia and systemic metabolic remodeling during muscle mitochondrial stress" and decided to address this point in the discussion (see lines 296-306) where it becomes clearer what we mean by that.
2. Figures 2, 3 and 4 miss to include outstanding data from the GDF15 KO mouse, and this is essential to validate some of the major conclusions of the manuscript. Only Figure 4 is complemented by some of those data included as Figure S1. Based on this, it is unclear at this moment, whether ablation of GDF15 is important or not for the effects of UCP1 expression on mitochondrial stress, muscle metabolism, adiposity, or browning of subcutaneous adipose tissue.

Response: Point well taken. We had omitted the KO control data in most figures because we did not find significant differences between WT and Gdf15-KO (KO) in any of the parameters investigated and did not want to make the figures too crowded. But of course, this control group is of importance. We have now have added expanded data figures (Fig EV1, EV3 – EV5) to each main figure where we show the comparisons between WT and KO separately.
3. The authors also document that UCP1 transgenic mice show an alteration in diurnal regulation of food intake. However, they at least should determine whether GDF15 is sufficient to induce these effects in mice. Does chronic GDF15 treatment to normal mice reduce in diurnal food

intake?

Response: We highly appreciate this point. To our best knowledge, we here provide the first evidence of a daytime-restricted anorectic action of endogenous GDF15. Whether this diurnal variation occurs only in the context of chronic mitochondrial dysfunction or in other pathophysiological conditions when circulating GDF15 is high (such as cancer, cardiovascular disease or sepsis), is yet to be investigated. However, we show that the diurnal anorexia of *Ucp1*-TG mice is completely abolished when GDF15 is ablated. This clearly demonstrates that muscle mitochondrial stress-induced GDF15 drives this daytime-restricted suppression of food intake in *Ucp1*-TG mice.

Importantly, we did not mean to imply that in wild-type (WT) mice endogenous GDF15 is responsible for diurnal regulation of food intake. This is evident from the GDF15 KO mice which do not show any alterations in the diurnal pattern of food intake, energy expenditure or activity (Fig EV4).

Moreover, we agree that it is an intriguing question, whether chronic treatment of normal mice with GDF15 promotes a daytime-restricted reduction in food intake similar to what we observe in *Ucp1*-TG mice. The anorectic action of exogenously administered GDF15 in high doses has been well documented by different groups (Emmerson et al, 2017; Hsu et al, 2017; Mullican et al, 2017; Patel et al, 2019; Tsai et al, 2018). Pharmacological administration of GDF15 to healthy WT mice promotes supraphysiological plasma concentrations (10 to 100ng/ml with the first 4hrs after injection) compared to endogenous GDF15 induction observed in *Ucp1*-TG animals (0.5 to 1ng/ml). It was shown that a single subcutaneous injection of GDF15 given immediately prior to the onset of the dark cycle results in a corresponding dose-dependent reduction of food intake that only reaches statistical significance at the highest dose (0.1 mg/kg; up to 100ng/ml GDF15 plasma concentration), whereas low (0.001 mg/kg) and mid (0.01 mg/kg) doses of GDF15 injection leading to plasma concentrations of 0.5 to 5 ng/ml were unable to suppress food intake (Patel et al, 2019). This suggests that only supraphysiological plasma concentrations of GDF15 are able to promote anorexia at night in WT mice. In rats, it was further shown that chronically subcutaneous injection of GDF15 (4nmol/kg) at 1–2 hours before the dark phase strongly suppresses 24hr food intake (Yang et al, 2017). However, none of the above mentioned studies performed a comprehensive 24hr metabolic profiling to investigate diurnal variations of food intake. One possibility to address this issue could be to treat healthy WT mice (acutely and chronically) with GDF15 supplied by osmotic mini pumps to obtain a stable, defined release and monitor food intake in a 24hr time course. This will be most interesting to address but would be beyond the scope of our study regarding the pathophysiological role of endogenous GDF15 in the context of muscle mitochondrial dysfunction and is certainly not feasible within 3 months of revision time.

Referee #2:

The manuscript by Ost and colleagues showed that a transgenic mouse with skeletal muscle-specific mitochondrial OXPHOS defect (*Ucp1*-TG), had induction of diurnal variation of GDF15. Knockout of GDF-15 in the *Ucp1*-TG promoted a progressive increase of body fat mass, which was associated with suppression of day time food intake resembling a pseudo-intermittent fasting. This led to adipose tissue remodeling, hypoinsulinemia, and increased energy metabolism. They conclude that GDF15 has an important role in diurnal regulation of energy balance during mitochondrial dysfunction. They also showed that sustained GDF15 elevation is not protective for muscle.

The study is of high quality and the use of genetically modified mice very appropriate to address the scientific question. The manuscript has two main messages. The first one is quite convincing, showing adipose tissue metabolic remodeling blunted in adult TGxKO mice. However, the second one, related to circadian changes, the data was not as strong. I could not really see the elevated carbohydrate oxidation during day time and increased lipid oxidation at night in TG mice, abolished in TGxKO mice (Fig 4C and D). In fact, a lot of the diurnal differences were quite minimal, and one wonders how physiologically relevant they are. It is possible that I am not interpreting the data correctly, so I would be glad to let the authors argue their case.

Response: Thank you for your comments and suggestions. We appreciate that you find our study of high quality and the data on adipose tissue remodeling convincing. Regarding the circadian changes, we tried to make this clearer by modifying the former Fig. 4 and 5 and presenting the data in different ways. The most important read-out concerning the diurnal changes of substrate oxidation is

the respiratory quotient (RQ) which is reflective of substrate oxidation (see Fig. 5J-L). The RQ refers to the quantity of CO₂ produced in relation to O₂ consumed. This ratio is 1 when glucose is oxidized and 0.7 when fat (assuming an average fat) is oxidized. For oxidation of protein (with an average amino acid composition) the RQ is around 0.8 (Jeukendrup & Wallis, 2005). The RQ thus reflects overall substrate utilization very precisely which is also evident by the very small variation (error bars). The carbohydrate and lipid oxidation were calculated from the RQ data (assuming a constant protein oxidation) and are thus not independently derived data. Therefore, for clarity reasons we decided to omit the calculated data on substrate respiration and present the actual measured data only. We also restructured the Fig. 4 (now Fig 5) to make the changes in diurnal and daily energy balance better visible (see also our response to referee #3, comment 4).

Referee #3:

In this manuscript by Ost et al., the authors provide evidence that GDF15 is produced in muscle under conditions of chronic mitochondrial uncoupling (overexpression of UCP1) that has mostly peripheral effects (i.e. not in muscle), leading them to conclude muscle-derived GDF15 is a mitokine. Arguably, the most interesting of these peripheral effects is the demonstration of GDF15-dependent suppression of day time food intake (i.e. circadian effects). The study is quite well executed and the primary data are robust. However, there are some concerns with experimental design and other issues that undercut the very strong conclusion reached that need to be addressed. These are detailed below:

Response: Thank you for your encouraging comments and suggestions. We tried to address all your specific comments and to strengthen our data by performing additional experiments and analyses.

1. The overall study design precludes an unequivocal conclusion about muscle being the primary source of the active Gdf15. That is, even though the mitochondrial stress is in muscle due to the muscle-specific HSA driver (although expression in esophagus and diaphragm should be discussed, McCarthy et al. 2012), the GDF15 KO is global. Thus, the possibility exists that muscle mitochondrial stress is triggering GDF15 expression/secretion elsewhere. Obviously, in this regard, knocking out GDF15 only in muscle is the key experiment. Furthermore, it is also possible that loss of Gdf15 expression in other tissues contributes to the phenotypes observed in the KO and TGxKO mice. The eWAT is a highly responsive fat depot (to diet, hormonal regulation and other stressors). Thus, it would be interesting to see if Gdf15 is expressed here, and if it is released from the eWAT.

Response: Point taken well. Thank you for pointing out the paper by McCarthy et al. regarding the HSA-construct. This paper shows that the HSA promoter drives expression in different striated muscles (including diaphragm, esophagus and tongue) but not in heart and smooth muscle (McCarthy et al, 2012). In our analyses we focused on quadriceps/gastrocnemius muscle because they are representative of mixed fiber skeletal muscle. We did not mean to imply that transgenic GDF15 is expressed and secreted just from these particular muscles investigated. However, the McCarthy paper clearly shows that the transgenic construct is expressed in striated muscle only. To confirm the mitochondrial uncoupling driven expression of GDF15, we analyzed GDF15 gene expression additionally in different muscle types (EDL, tibialis anterior, gastrocnemius, quadriceps, soleus, diaphragm, and esophagus) compared to heart, liver, kidney, spleen, lung and different adipose tissue depots. As evident from the heatmap (Fig. 1F) *Gdf15* expression is very low in WT mixed or glycolytic muscle (ct values around and above 30), somewhat higher in WT oxidative muscles (soleus and diaphragm, ct values 26-27) with similar expression levels as in liver and sWAT, and lowest in iBAT. In *Ucp1*-TG mice there was a clear increase in expression only in skeletal muscle i.e. in striated muscle tissues where the HSA-driven *Ucp1*-transgene is expressed (Klaus et al, 2005). In all other tissues including eWAT we could not detect any significant differences in *Gdf15* gene expression between WT and *Ucp1*-TG mice (Fig. 1G). In addition, we analyzed GDF15 protein levels in muscle tissue and ex vivo secretion from muscle (Fig. 1H, I) which showed that GDF15 protein was not detectable in WT muscle whereas it was detectable in and secreted from soleus and EDL muscle of *Ucp1*-TG mice. Additional in vitro experiments with C2C12 myocytes (Fig. 1M,N) show the dose-dependent induction of the ISR linked to increased GDF15 expression by mitochondrial uncoupling (treatment with FCCP). Analyses of day/night expression of GDF15 confirmed that only in muscle we could find increased gene expression

of GDF15 in *Ucp1*-Tg mice (Fig. 6E) and that their muscle protein levels reflect circulating GDF15 (Fig. 6 F, G). In our opinion, this shows that the increase in circulating GDF15 is due to the mitochondrial stress dependent, muscle specific increase in *Gdf15* gene expression. As shown in the extended Fig. EV1, EV2-EV4, global GDF15 ablation (*Gdf15*-KO) had basically no effect on any of the measured variables which shows that basal, circulating GDF15 does not play a metabolic role in context with the physiological functions investigated in our study. Therefore, we think it is reasonable to assume that a muscle-specific *Gdf15* knockout would not show a different phenotype compared to the global *Gdf15* knockout during muscle mitochondrial dysfunction.

2. Although the data presented suggest that there is minimal effect of increased *Gdf15* muscle expression on the muscle itself, it is a stretch for the authors to conclude that there is no direct muscle-adipose tissue crosstalk (line 160-161). For example, UCP1 protein expression in sWAT was apparently completely decreased in TGxKO mice (Fig. 3D) and the mRNA levels of several markers of "browning" were decreased in sWAT of TGxKO. Thus, how can the authors conclude that there was no impact of *Gdf15* loss on sWAT? Additional measurements are needed in this regard: for example, quantification of regulated circulating factors that remodel adipose and could contribute to the "browning," effects of GDF15 on eWAT depots, and measurement of WAT differentiation markers expression like PPAR α .

Response: We apologize if our wording was not clear apparently leading to a misunderstanding. Of course, we cannot exclude a direct effect of GDF15 on WAT. The loss of GDF15 definitely affects sWAT morphology and browning phenotype as shown here. However, we believe that this is most probably a centrally mediated effect, since we could not detect expression of the GDF15 receptor (GFRAL) in any WAT depot. We now include new data on the eWAT depot and performed additional gene expression analyses. As shown in Fig. 1F and G, *Gdf15* gene expression in eWAT was very low and not induced in *Ucp1*-TG mice. However, similar to sWAT, eWAT weight was increased in TGxKO mice compared to *Ucp1*-TG (Fig. 3G, H). Otherwise, there were no differences in eWAT gene expression between any of the groups (Fig 4E). Of note, adipose tissue differentiation markers (PPARs) showed no differences in any of the adipose depots (Fig. 4B, E). Our data in Fig. 4A-D clearly show the loss of browning in sWAT in *Ucp1*-TG mice upon ablation of GDF15. The most effective endogenous browning factor is FGF21, and we have shown previously that the genetic loss of FGF21 completely inhibits browning of sWAT in *Ucp1*-TG mice (Ost et al, 2016). Interestingly, the increase in FGF21 observed in *Ucp1*-TG mice was not affected by loss of GDF15 (Fig. 4F) but sWAT browning was still suppressed. This shows that high circulating FGF21 alone is not sufficient to induce browning and suggests that GDF15 affects browning independently of FGF21. This could possibly be an indirect action due to the increased fat mass. This point has been added to the discussion (see lines 277 -295).

3. Given the extensive literature demonstrating that *Gdf15* regulates adiposity via central regulation of food intake, the authors should measure satiety hormones & peptides (e.g. CART, alpha-MSH, leptin) in the different mouse models and investigate whether *Gdf15* impacts their regulation in a circadian manner, and if loss of *Gdf15* alters levels or circadian expression of these factors.

Response: We have previously analyzed hypothalamic gene expression of neuropeptides involved in appetite control (MC4R, NPY, PMCH, and POMC) and could not detect any differences between *Ucp1*-TG and WT mice (unpublished data), suggesting that these pathways are likely not involved in the GDF15 action. Unfortunately, we did not collect brain/hypothalamus samples in the day/night experiment to repeat these measurements. In Fig 4G we show that loss of GDF15 in the *Ucp1*-TG mice increases circulating leptin levels even higher than in WT controls. Measurement of circulating leptin levels in the day/night experiment did not show major differences between WT and TG mice (data not included in the revised manuscript).

4. The authors demonstrate that there is no significant difference in energy intake (Fig. 4H) or energy expenditure (Fig. 4K) between TG-KO and WT mice. However, there appears to be a consistently higher rate of fat mass accumulation in the TG-KO mice relative to the WT mice (Fig. 3A-C). What was the statistical significance for differences between TG-KO and WT for the fat mass data? How do the authors reconcile the lack of any apparent difference in food intake between WT and TG-KO with the relatively higher fat mass accumulation in the TG-

KO mice? The authors state in line 198-199 that the increased daytime energy intake "can completely explain the progressive fat accumulation observed in TGxKO mice." This is a very aggressive statement and the data presented in the manuscript do not provide enough evidence to make this conclusion. The authors should consider subjecting the TGxKO, WT and TG mice to a pair-feeding regimen to determine whether equalizing energy intake across all groups will reverse the differences in fat mass accumulation.

Response: We apologize for this ambiguity. Body fat data (including statistical analysis) are shown in Fig 3C + 3F. One problem when looking at energy balance is that small daily changes (that hardly reach statistical significance) accumulate over time and can result in significant changes of fat mass in the long term. We measured energy assimilation and energy expenditure in week 17 which (although not statistically significant) resulted in 3.5 kJ daily difference in energy balance between TGxKO and WT mice (Fig. 5I). Assuming that the energy content of fat mass is 39 kJ/g (if all of this were lipids) a 3.5 kJ daily energy surplus amounts to 24.5 kJ per week which corresponds to 0.63 g fat mass and in 10 weeks to over 6.3 g differences in fat mass accrual. Of course this is theoretical and in reality body fat development is not linear over time. But importantly, the fat mass difference between WT and TGxKO mice of 1.9 g (week 20) and 4.8 g (week 45), respectively, can be completely accounted for by the difference in energy balance. This is what we intended to express by our statement. We tried to rephrase it to make it better understandable (see lines 209 – 222). Since the difference in energy balance is more than sufficient to explain the fat mass differences between WT, TG, and TGxKO mice we do not think it be appropriate here to perform a pair feeding experiment. Pair feeding is also problematic in a way that it disrupts the diurnal feeding rhythm which in itself can affect energy metabolism. If offered a limited amount of food only, mice tend to consume it rapidly resulting in a prolonged daily fasting period compared to ad libitum feeding. For their Nature Medicine article from 2017, Yang et al. performed a pair feeding study to assess the mechanisms driving GDF15 mediated weight loss and came to the conclusion that body weight reduction by GDF15 is mediated solely through the central suppression of food intake (Yang et al, 2017) which is perfectly in line with our data.

5. The transcriptional regulation of *Gdf15* expression by mitochondrial dysfunction, or specifically uncoupling in the UCP1-TG mice, could have been investigated in greater detail. How does mitochondrial uncoupling lead to up-regulation of *Gdf15* expression? What are the pathways involved, and what transcriptional regulators are recruited to increase activity at the *Gdf15* promoter? The authors suggest (lines 90-91) that a CHOP-dependent mechanism of *Gdf15* upregulation might explain the increased *Gdf15* observed with mitochondrial uncoupling in the TG mice. However, in Fig. 5 the data clearly suggests that even though expression of CHOP and other ISR mediators is maintained day or night, *Gdf15* expression and plasma levels fluctuate significantly. How do the authors explain this?

Response: This point is well taken. The induction of GDF15 in different cell types by the integrated stress response (ISR) has been explored in detail (Chung et al, 2017; Patel et al, 2019) which is in line with our own previous data (Ost et al, 2015) and the results presented in Fig 1N. Chung et al. performed detailed analysis of transcription factors involved in GDF15 induction in response to mitochondrial stress in skeletal muscle. They reported that stress induced GDF15 induction in muscle was CHOP dependent (Chung et al, 2017) which fits very well with our data reported (Fig. 1C, N). In our study we thus focused on the metabolic effects of GDF15. Indeed, we did not find diurnal differences in the expression of ISR genes (Fig. 6A) or eIF2 alpha phosphorylation (Fig. 6B + C) which is in line with the fact that UCP1 expression also did not show diurnal variations (data not shown in the revised manuscript). Tissue specific circadian regulation is apparently not affected in *Ucp1*-TG mice as evident from the maintained diurnal gene expression of the transcriptional repressor *Rev-Erb alpha*, an important circadian regulator (Fig. 6D). Hence, there must be other factors driving the diurnal variation in GDF15 mRNA levels, possibly by affecting RNA stability and translation which need further exploration. This point has been added to the discussion (see lines 310 - 313).

6. Does the circadian rhythmicity of *Gdf15* expression alter as the animal ages? It would be interesting to see if there is a difference in muscle expression, plasma levels, and in the delta change between day/night, as the animal ages.

Response: This is indeed an interesting question. As shown in Fig 1L, GDF15 levels were not further increased in old *Ucp1*-TG mice compared to young mice which, however, do not

exclude a possible change in diurnal GDF15 expression in old mice. Unfortunately, we currently have no *Ucp1*-TG mice old enough to perform these analyses. We are breeding these mice but it will take several more months until we will have old mice available for further studies.

7. Does HFD feeding impact the circadian rhythmicity of *Gdf15* expression? Are there diurnal variations in *Gdf15* expression in other tissues?

Response: We appreciate the point. We have shown previously that muscle *Gdf15* gene expression is similar in low and high fat fed *Ucp1*-Tg mice (Ost et al, 2016) and also see no differences in plasma levels (Fig. 1L) suggesting that mitochondrial stress induced GDF15 is not affected by HFD feeding. Nevertheless, it is an interesting question which could be addressed in future studies. We checked for diurnal variations of *Gdf15* expression in liver and eWAT and found a tendency for a diurnal regulation in liver, both in WT and TG mice, but no evidence of a diurnal regulation in eWAT which shows very low mRNA levels anyway (Fig 6E).

8. The Introduction and Discussion could be written more succinctly and clearly.

Response: We rewrote the introduction and discussion for more successiveness and clarity.

Minor points

- > Line 34: "manifest at any age and organ ..."; meaning is unclear
- > Line 39: "thereby described firstly ..."; awkward
- > Line 50: "secreted molecules" vague, please use mitokine/cytokine/hormone
- > Line 70: "activating adipose tissue" vague; what is meant by activating?
- > Use of acronyms (especially the name of the mouse *Ucp1*-TG): once an acronym has been introduced and the full written form has been indicated, the acronym only should be used for all subsequent mentions (no need to repeatedly use the full form).
- > The acronym mtISR is introduced at line 91 as a specific type of ISR, but later mentions of ISR are not specified at mtISR (eg. line 136, 208). Why did the authors specify mtISR, and is it necessary? The authors should decide how to reference either ISR or mtISR through the text and evaluate the need for two acronyms.
- > Line 204: Were the times for day and night correctly stated? As written in the manuscript, the day phase and night phase overlap.

Response: Thank you for pointing these out, we have corrected the text accordingly.

References cited:

Chung HK, Ryu D, Kim KS, Chang JY, Kim YK, Yi HS, Kang SG, Choi MJ, Lee SE, Jung SB et al (2017) Growth differentiation factor 15 is a myomitokine governing systemic energy homeostasis. *J Cell Biol* 216: 149-165

Emmerson PJ, Wang F, Du Y, Liu Q, Pickard RT, Gonciarz MD, Coskun T, Hamang MJ, Sindelar DK, Ballman KK et al (2017) The metabolic effects of GDF15 are mediated by the orphan receptor GFRAL. *Nat Med* 23: 1215-1219

Hsu JY, Crawley S, Chen M, Ayupova DA, Lindhout DA, Higbee J, Kutach A, Joo W, Gao Z, Fu D et al (2017) Non-homeostatic body weight regulation through a brainstem-restricted receptor for GDF15. *Nature* 550: 255-259

Jeukendrup AE, Wallis GA (2005) Measurement of substrate oxidation during exercise by means of gas exchange measurements. *Int J Sports Med* 26 Suppl 1: S28-37

Klaus S, Rudolph B, Dohrmann C, Wehr R (2005) Expression of uncoupling protein 1 in skeletal muscle decreases muscle energy efficiency and affects thermoregulation and substrate oxidation. *Physiol Genomics* 21: 193-200

McCarthy JJ, Srikuea R, Kirby TJ, Peterson CA, Esser KA (2012) Inducible Cre transgenic mouse strain for skeletal muscle-specific gene targeting. *Skelet Muscle* 2: 8

Mullican SE, Lin-Schmidt X, Chin C-N, Chavez JA, Furman JL, Armstrong AA, Beck SC, South VJ, Dinh TQ, Cash-Mason TD (2017) GFRAL is the receptor for GDF15 and the ligand promotes weight loss in mice and nonhuman primates. *Nature medicine* 23: 1150

Ost M, Coleman V, Voigt A, van Schothorst EM, Keipert S, van der Stelt I, Ringel S, Graja A, Ambrosi T, Kipp AP et al (2016) Muscle mitochondrial stress adaptation operates independently of endogenous FGF21 action. *Mol Metab* 5: 79-90

Ost M, Keipert S, van Schothorst EM, Donner V, van der Stelt I, Kipp AP, Petzke KJ, Jove M, Pamplona R, Portero-Otin M et al (2015) Muscle mitohormesis promotes cellular survival via serine/glycine pathway flux. *FASEB journal* 29: 1314-1328

Patel S, Alvarez-Guaita A, Melvin A, Rimmington D, Dattilo A, Miedzybrodzka EL, Cimino I, Maurin AC, Roberts GP, Meek CL et al (2019) GDF15 Provides an Endocrine Signal of Nutritional Stress in Mice and Humans. *Cell Metab* 29: 707-718 e708

Tsai VW, Zhang HP, Manandhar R, Lee-Ng KKM, Lebhar H, Marquis CP, Husaini Y, Sainsbury A, Brown DA, Breit SN (2018) Treatment with the TGF- β superfamily cytokine MIC-1/GDF15 reduces the adiposity and corrects the metabolic dysfunction of mice with diet-induced obesity. *Int J Obes (Lond)* 42: 561-571

Yang L, Chang CC, Sun Z, Madsen D, Zhu H, Padkjaer SB, Wu X, Huang T, Hultman K, Paulsen SJ et al (2017) GFRAL is the receptor for GDF15 and is required for the anti-obesity effects of the ligand. *Nat Med* 23: 1158-1166

2nd Editorial Decision

9 December 2019

Thank you for submitting the revised version of your manuscript. It has now been seen by all of the original referees.

As you can see, the referees find that the study is significantly improved during revision and recommend publication here. Before I can accept the manuscript, I need you to address some minor points below:

- Please address the remaining minor concerns of referee #3.

•

REFeree REPORTS

Referee #1:

All my previous comments have been adequately answered and I think the manuscript has the quality to be published in EMBO Rep.

Referee #2:

The authors have addressed the Reviewers' concerns. I am satisfied.

Referee #3:

The authors have provided new data, explanations and rewrites that have improved the manuscript considerably. Furthermore, we agree that some of our initial comments (e.g. effects of age, detailed mechanisms of transcriptional regulation, etc.) could be considered beyond the scope of the current study. That said, a few remaining points need to be addressed based on the authors responses and new data provided.

- With regard to the authors response to our original point #2. First, while the authors have included important new data showing that there is no GDF15 gene expression induced in the eWAT of TG mice relative to WT, this does not address the possibility that increases in circulating GDF15 may impact the sWAT depot. It is important that the authors measured the GDF15 receptor GFRAL in the eWAT and sWAT depots and did not detect expression in these depots but only in the brain. These data should be included in manuscript with appropriate positive controls. Second, the authors mention FGF21 as a possible critical mediator of the browning effects seen in the TG mice. However, they also note that browning of the sWAT is lost in the TG-UCP with GDF15 KO, but FGF21 levels are unchanged. While the authors do discuss this further in the revised manuscript (lines 277-295), their final conclusion (line 292-293) is that "GDF15 affects browning independently of FGF21, possibly due to an indirect action fat mass expansion". What is meant by "indirect action fat mass expansion"? The authors need to state exactly what they mean here and provide a better interpretation of their GDF15 results with regard to browning vis-à-vis known effects of FGF21.

- With regard to the authors response to our original point #3. The data in Figure 4G demonstrate significant increases in circulating leptin in the TGxKO mice. How do the authors explain that there is no significant impact on food intake, when both GDF15 and leptin have major effects on food intake based on multiple reports in the literature? This needs to be discussed more directly in the manuscript.

2nd Revision - authors' response

16 December 2019

REFEREE COMMENTS**Referee #1:**

All my previous comments have been adequately answered and I think the manuscript has the quality to be published in EMBO Rep. **Response:** Thanks for appreciating our work.

Referee #2:

The authors have addressed the Reviewers' concerns. I am satisfied. **Response:** Thanks for appreciating our work.

Referee #3:

The authors have provided new data, explanations and rewrites that have improved the manuscript considerably. Furthermore, we agree that some of our initial comments (e.g. effects of age, detailed mechanisms of transcriptional regulation, etc.) could be considered beyond the scope of the current study. That said, a few remaining points need to be addressed based on the authors responses and new data provided.

1. With regard to the authors response to our original point #2. First, while the authors have included important new data showing that there is no GDF15 gene expression induced in the eWAT of TG mice relative to WT, this does not address the possibility that increases in circulating GDF15 may impact the sWAT depot. It is important that the authors measured the GDF15 receptor GFRAL in the eWAT and sWAT depots and did not detect expression

in these depots but only in the brain. These data should be included in manuscript with appropriate positive controls.

Response: Point well taken. In our study, we were not able to detect any *Gfrol* mRNA expression in sWAT or eWAT of WT, KO, TG or TGxKO mice (data not included in the manuscript). We were using PCR primers designed by our group (Primer3 software) as well as the same PCR primers used in the study of (Mullican et al., 2017). Importantly, an adequate positive control would require a selective collection of hindbrain tissue samples, in particular area postrema and nucleus tractus solitarius (NST). However, we only collected brain tissue for histological analysis and tissue for RNA isolation would require additional animals. Moreover, it has been demonstrated multiple times before, that GFRAL is not expressed in WAT depots (Emmerson et al., 2017; Hsu et al., 2017; Li et al., 2005; Luan et al., 2019; Mullican et al., 2017; Yang et al., 2017). Thus, we decided to address this point in the discussion (see lines 275-278).

2. Second, the authors mention FGF21 as a possible critical mediator of the browning effects seen in the TG mice. However, they also note that browning of the sWAT is lost in the TG-UCP with GDF15 KO, but FGF21 levels are unchanged. While the authors do discuss this further in the revised manuscript (lines 277-295), their final conclusion (line 292-293) is that "GDF15 affects browning independently of FGF21, possibly due to an indirect action fat mass expansion". What is meant by "indirect action fat mass expansion"? The authors need to state exactly what they mean here and provide a better interpretation of their GDF15 results with regard to browning vis-à-vis known effects of FGF21.

Response: We apologize if our wording was not clear apparently leading to a misunderstanding. We noticed a semantic error: we meant that the lack of GDF15 affects browning of sWAT possibly due to an indirect effect via fat mass expansion. Notably, it has been described before, that adiposity in mice promotes a suppression of WAT browning (Geurts et al., 2015) and coordinates a restructuring of metabolism that could contribute to the whitening of adipose tissue (Cummins et al., 2014). Here, our data in Fig. 4A-D clearly show the loss of browning in sWAT upon ablation of GDF15 in TGxKO mice together with an increased adiposity (Fig. 3 + 4). As mentioned before, the most effective endogenous browning factor is FGF21, and we have shown previously that the genetic loss of FGF21 completely inhibits browning of sWAT in *Ucp1*-TG mice (Ost et al., 2016). However, the increase in FGF21 observed in TG mice was not affected by loss of GDF15 (Fig. 4F) but sWAT browning was still suppressed. This shows that high circulating FGF21 alone is not sufficient to induce browning during muscle mitochondrial dysfunction. Thus, we believe that the GDF15-dependent loss of sWAT browning could possibly be an indirect action due to the progressive fat mass expansion of TGxKO mice. This is now adapted in the revised manuscript (see lines 287 – 293).

3. With regard to the authors response to our original point #3. The data in Figure 4G demonstrate significant increases in circulating leptin in the TGxKO mice. How do the authors explain that there is no significant impact on food intake, when both GDF15 and leptin have major effects on food intake based on multiple reports in the literature? This needs to be discussed more directly in the manuscript.

Response: We highly appreciate this point. It is well-accepted that plasma levels of leptin are reflected by the amount of fat mass, meaning they are increased in obesity (Blum et al., 1997; Geurts et al., 2015; Ostlund et al., 1996) which is commonly interpreted as a leptin resistance. In line with that, with the increased adiposity we here observed higher levels of circulating leptin in TGxKO mice, suggesting no particular disturbance of the leptin axis in the TG mouse model (see lines 293 – 295). It remains to be elucidated whether this already reflects a leptin resistant state, but we believe that this is beyond the scope of our current study.

References:

Blum, W.F., Englaro, P., Hanitsch, S., Juul, A., Hertel, N.T., Muller, J., Skakkebaek, N.E., Heiman, M.L., Birkett, M., Attanasio, A.M., et al. (1997). Plasma leptin levels in healthy children and adolescents: dependence on body mass index, body fat mass, gender, pubertal stage, and testosterone. *J Clin Endocrinol Metab* 82, 2904-2910.

Cummins, T.D., Holden, C.R., Sansbury, B.E., Gibb, A.A., Shah, J., Zafar, N., Tang, Y., Hellmann, J., Rai, S.N., Spite, M., et al. (2014). Metabolic remodeling of white adipose tissue in obesity. *Am J Physiol Endocrinol Metab* 307, E262-277.

Emmerson, P.J., Wang, F., Du, Y., Liu, Q., Pickard, R.T., Gonciarz, M.D., Coskun, T., Hamang, M.J., Sindelar, D.K., Ballman, K.K., et al. (2017). The metabolic effects of GDF15 are mediated by the orphan receptor GFRAL. *Nat Med* 23, 1215-1219.

Geurts, L., Everard, A., Van Hul, M., Essaghir, A., Duparc, T., Matamoros, S., Plovier, H., Castel, J., Denis, R.G., Bergiers, M., et al. (2015). Adipose tissue NAPE-PLD controls fat mass development by altering the browning process and gut microbiota. *Nat Commun* 6, 6495.

Hsu, J.Y., Crawley, S., Chen, M., Ayupova, D.A., Lindhout, D.A., Higbee, J., Kutach, A., Joo, W., Gao, Z., Fu, D., et al. (2017). Non-homeostatic body weight regulation through a brainstem-restricted receptor for GDF15. *Nature* 550, 255-259.

Li, Z., Wang, B., Wu, X., Cheng, S.Y., Paraoan, L., and Zhou, J. (2005). Identification, expression and functional characterization of the GRAL gene. *J Neurochem* 95, 361-376.

Luan, H.H., Wang, A., Hilliard, B.K., Carvalho, F., Rosen, C.E., Ahasic, A.M., Herzog, E.L., Kang, I., Pisani, M.A., Yu, S., et al. (2019). GDF15 Is an Inflammation-Induced Central Mediator of Tissue Tolerance. *Cell* 178, 1231-1244 e1211.

Mullican, S.E., Lin-Schmidt, X., Chin, C.N., Chavez, J.A., Furman, J.L., Armstrong, A.A., Beck, S.C., South, V.J., Dinh, T.Q., Cash-Mason, T.D., et al. (2017). GFRAL is the receptor for GDF15 and the ligand promotes weight loss in mice and nonhuman primates. *Nat Med* 23, 1150-1157.

Ost, M., Coleman, V., Voigt, A., van Schothorst, E.M., Keipert, S., van der Stelt, I., Ringel, S., Graja, A., Ambrosi, T., Kipp, A.P., et al. (2016). Muscle mitochondrial stress adaptation operates independently of endogenous FGF21 action. *Mol Metab* 5, 79-90.

Ostlund, R.E., Jr., Yang, J.W., Klein, S., and Gingerich, R. (1996). Relation between plasma leptin concentration and body fat, gender, diet, age, and metabolic covariates. *J Clin Endocrinol Metab* 81, 3909-3913.

Yang, L., Chang, C.C., Sun, Z., Madsen, D., Zhu, H., Padkjaer, S.B., Wu, X., Huang, T., Hultman, K., Paulsen, S.J., et al. (2017). GFRAL is the receptor for GDF15 and is required for the anti-obesity effects of the ligand. *Nat Med* 23, 1158-1166.

Accepted

10 January 2020

Thank you for submitting your revised manuscript. I have now looked at everything and all looks fine. Therefore I am very pleased to accept your manuscript for publication in EMBO Reports.

Corresponding Author Name: Mario Ost, Susanne Klaus

Manuscript Number: EMBOR-2019-48804